# Seasonal and diurnal freeze-thaw dynamics of a rock glacier and their impacts on mixing and solute transport

Cyprien Louis[1], Landon J.S. Halloran[1,*], and Clément Roques[1,*]

[1]Centre for Hydrogeology and Geothermics (CHYN), University of Neuchâtel, rue Emile-Argand 11, 2000 Neuchâtel, Switzerland

[*]These authors contributed equally to this work.

**Correspondence:** Landon J.S. Halloran (landon.halloran@unine.ch) and Clément Roques (clement.roques@unine.ch)

**Abstract.** Rock glaciers play a vital role in the hydrological functioning of many alpine catchments. Here, we investigate seasonal and daily freeze-thaw cycles of the previously undocumented Canfinal rock glacier (RG) located in the Val d'Ursé catchment (Bernina Range, Switzerland) and the RG's influence on the dynamics of the hydrogeological system. We combine digital image correlation techniques, geochemical and isotopic analyses, time-series analysis, and hydrological monitoring to understand the functioning of the hydrological system. An acceleration of RG creep since 1990 has occurred, with the most active regions exhibiting horizontal velocities of ∼1 m/yr. Distinct geochemical signatures of springs influenced by RG discharge reflect contrasting and temporally-variable groundwater mixing ratios. A novel application of frequency-domain analysis to time-series of air temperature and spring electrical conductivity enables a quantitative understanding of the RG thaw and subsurface flow dynamics. Following the onset of snowmelt, we observed a gradual decrease in the time-lag between air temperature maxima and spring electrical conductivity minima at the front of the rock glacier. This suggests progressively increasing flows within the talus, driven by efficient recharge from snowmelt and contribution from the thawing rock glacier. Through our multi-method approach, we develop a conceptual models representing the main cryo-hydrogeological processes involved in RG-influenced alpine headwaters.

## 1 Introduction

Recent studies have quantified the role of groundwater in sustaining perennial streamflow in both glaciated and non-glaciated alpine headwater catchments (Somers et al., 2019; Halloran et al., 2023). Rock glaciers (RGs) are key features in alpine headwaters, being the most common periglacial phenomena on Earth (Krainer et al., 2007). They can store significant volumes of water as groundwater and ice and contribute to catchment-scale fluxes (Harris et al., 2009; Evans et al., 2018). Rising temperatures and changes in precipitation are accelerating the thawing of RGs worldwide (Kääb et al., 2007; Marcer et al., 2021; Frei et al., 2018; Hanus et al., 2021; Manchado et al., 2024). Rock glacier degradation is expected to have a significant impacts on the alpine hydrological cycle, affecting the hydrology of downstream springs and streams (Jones et al., 2021).

The active layer has been shown to act as a thermal insulator between the surface and the frozen RG core (Humlum, 1997; Amschwand et al., 2023). It has a low thermal conductivity, which implies a different thermal regime between the RG and its surrounding environment. The ice is therefore partially insulated from external conditions, and thus thermally

decoupled, making it more resilient to climate change than glaciers (Anderson et al., 2018; Jones et al., 2019, 2021). Analysis of displacement dynamics can aid in tracking the seasonal freeze-thaw dynamics of active RGs (Groh and Blöthe, 2019). The creep of active RGs is primarily caused by forces at the primary basal shear horizon (Krainer et al., 2015). The dynamics of RGs change over time and at different scales depending on the evolution of the RG thermal regime. It is mainly controlled by heat conduction driven by air temperature, but other processes such as advection of air or water through the RG matrix can also play a role (Haeberli et al., 2006; Pruessner et al., 2018). Several studies have found an increase in horizontal velocities as a result of rising temperatures in the Alps due to climate change (Kääb et al., 2007; Marcer et al., 2021).

The critical link between RG freeze-thaw cycles, hydrological regimes, and creep dynamics is an underdeveloped topic (van Tiel et al.; Harrington et al., 2018; Pauritsch et al.). In the context of climate change, several hypotheses–which are not mutually-exclusive–have been proposed to explain modifications in the dynamics of horizontal displacement velocities observed in active RGs:

- In summer, high average and extreme air temperature values favour the melting of RG ice, leading to destabilisation and acceleration (Marcer et al., 2021).

- Since adequate snow cover acts as a thermal insulator for the ground, a colder, snowier winter will tend to protect the active zone of the RG in early spring (DallAmico et al., 2008).

- The early winter snow cover is important because it influences the thermal conditions underground by insulating the ground from the arrival of colder air. As a result, it is unfavourable for the preservation and formation of ice within the RG (Zhang, 2005).

- Intense precipitation in the form of rain has a destabilising effect due to the amount of water it can carry and the heat transfer it induces (Wirz et al., 2016).

Determining the long- and short-term dynamics of RG freeze-thaw cycles remains a major challenge (Pruessner et al., 2021). This knowledge gap stems from difficulties in collecting relevant data at such elevations, as well as a lack of understanding of the processes involved over pertinent time-scales. Difficulties lie in distinguishing between water fluxes from RGs and those from deeper groundwater flow paths, as well as in capturing their seasonal to daily variability (Krainer et al., 2015; Harrington et al., 2018) is challenging. Few studies that place RGs in their broader hydrogeological context have been conducted (e.g. Krainer et al., 2007; Del Siro et al., 2023; Munroe and Handwerger, 2023). As a result, seasonal variations in discharge sources such as groundwater, precipitation and snowmelt are often overlooked.

Environmental tracers provide unique information on the origin of waters, their residence times, and mixing during hydrological changes. Studies have used electrical conductivity (EC) as a tracer to complement water level data at alpine sites, taking advantage of its low cost and low probe maintenance requirements (Cano-Paoli et al., 2019). For example, EC can be used to separate hydrographs (Laudon and Slaymaker, 1997) or to analyse daily dilution/enrichment cycles of RG runoff on a seasonal or daily scale (Brighenti et al., 2021). EC is generally low at the onset of snowmelt and during high runoff periods due to significant dilution. In contrast, during late summer and winter baseflow, EC progressively increases as groundwater, with its

higher mineralization, becomes the primary driver of streamflow (Jones et al., 2019). Field campaigns for water sampling, on the other hand, face significant logistical challenges in alpine environments, limiting the ability to intervene at appropriate intervals to capture the variability of responses during high-frequency hydrological changes. Several authors have characterised the seasonal evolution of the composition of water discharge from RGs using isotopic analyses (in particular $^{18}$O). These studies suggest that only a small fraction of the water released by rock glaciers originates from ice thaw, with precipitation, snowmelt, and groundwater being the primary sources of spring water (Krainer et al., 2007, 2015; Harrington et al., 2018).

In this study, we employ a multi-method approach to investigate the cryo-hydrogeological functioning of the Val d'Ursé Critical Zone Observatory in Switzerland, where a major rock glacier (the Canfinal rock glacier) is present. By combining digital image correlation, hydrochemical analysis, and a novel frequency-domain analysis of temperature and electrical conductivity monitored at spring locations, we aim to resolve the seasonal to diurnal freeze-thaw dynamics of the RG, identify multi-year trends in its creep rates and their meteorological drivers, and elucidate cryosphere-groundwater interactions. This comprehensive approach not only enhances our understanding of the complex hydrological processes in the Val d'Ursé headwater catchment but also generates knowledge that can be applied to other alpine regions with similar environmental settings.

## 2   Study area and instrumentation

The Canfinal rock glacier is located in the UniNE Val d'Ursé Critical Zone Observatory (CZO) for Alpine Research (Grisons, Switzerland) (Figure 1). It is a glacially formed, E-W oriented valley located in the Bernina Range (Western Rhaetian Alps, Switzerland) that feeds the main Val Poschiavo valley to the east. The geology consists of crystalline basement rocks and intruded granite of the Sella Nappe. The lower and middle parts of the valley are concerned by Marinelli-Formation schist with Sella-Granodiorit (Lindsay, 2000). Musella-Granite and Sella-Granodiorit combine to form the southern ridges which feed the Canfinal RG. The Sella-Nappe is thrust over the Bernina-Nappe near the northern boundary of the Val d'Ursé and the nappe stack is part of a larger regional anticline.

Field outcrop observations reveal highly fractured and faulted lithologies, including open fractures, exfoliation joins, calcite veins, and folded schist (Lindsay, 2000). Analysis of fractures in borehole KB4 (Figure 1) indicates dip angles ranging from 30° to 90° from the horizontal, with fracture density decreasing with depth. This trend is consistent with findings from other crystalline regions (Eberhardt et al., 2004) and support classical conceptual structural models in alpine settings (Welch and Allen, 2014). This fracture pattern results from tectonic processes such as exhumation, stress release, and topographic stresses (Hencher et al., 2011). The main lineaments in the catchment are oriented NW-SE and NNE-SSW and are predominantly sub-vertical (Lindsay, 2000).

In the region, mean precipitation is 1060 mm over the period 2014-2022 based on measurements from nearby weather stations managed by the Federal Office of Meteorology and Climatology MeteoSwiss (Robbia, 1078 m.a.s.l and Passo del Bernina, 2272 m.a.s.l). Annual mean temperature measured at the front of the RG during 2014-2018 is 3.2 $^oC$. Typically in the Alps, peaks in monthly precipitation occur in early spring and late fall, while February is the driest and coldest month (Lindsay, 2000).

The hydrogeological system of the study site is made up of a fractured aquifer which is overlain by quaternary detrital sediments and coarse deposits. Several locations in the study area are intensely fractured, which allow for efficient deep groundwater recharge. Infiltration is expected to be greatest on gentle slopes with exposed permeable material such as debris and talus deposits or fractured bedrock. Several springs and groundwater seepage zones influenced by permafrost thaw have been identified across the watershed. They confirm the permafrost zones indicated by the Federal Office of Environment (Figure 1).

The Val d'Ursé CZO (Figure 1) includes a 200.3 m-deep borehole equipped with two pore water pressure sensors (KB4 in Figure 1). Additionally, the electrical conductivity (EC) and temperature of several springs (Figure 1) are monitored.

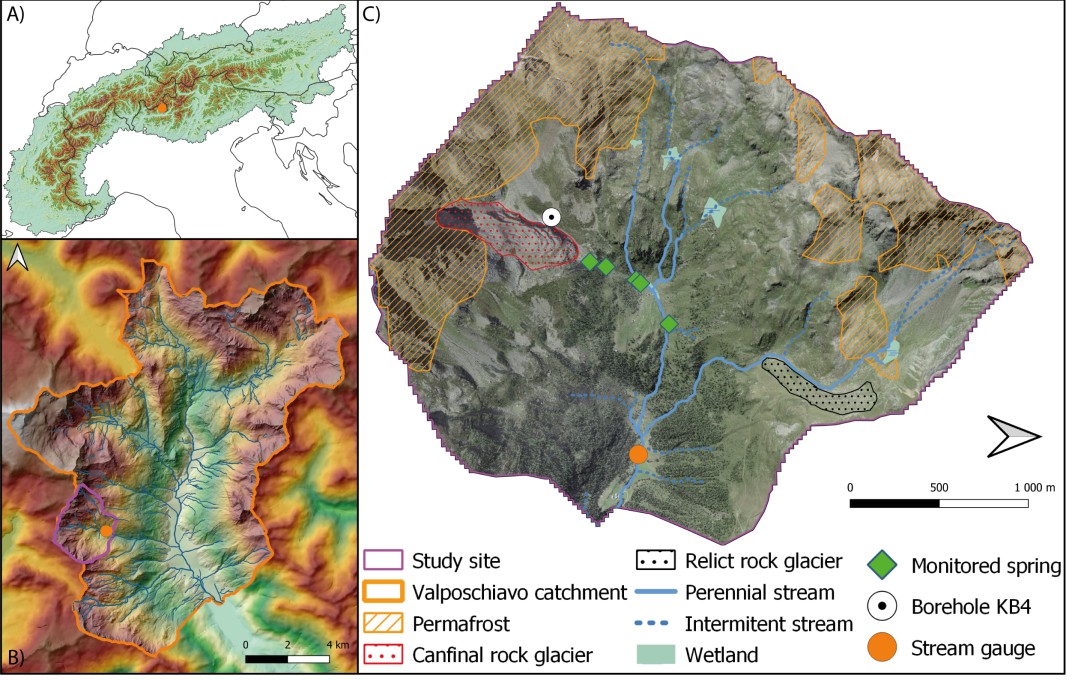

**Figure 1.** Location of the study site: *A)* in relation to the Alps, *B)* within the Poschiavino catchment, and *C)* the Val d'Ursé Critical Zone Observatory (CZO) for Alpine Research. The CZO is characterized by a dense network of springs, streams, wetlands, permafrost (FOEN Switzerland, 2005), and two major rock glaciers. It is supported by an extensive monitoring network, though only a selection of environmental monitoring points relevant to this study are shown.

A former glacier front, associated with the Cavaglia local glacial stage (Egesen I), reached an elevation of 2160 $m$ a.s.l. with an equilibrium line at ~2450 $m$ where the Canfinal RG now stands (Burga, 1987). Rockfalls from the cliffs filled the moraine valley of this retreated or partially retreated glacier, and favourable permafrost conditions led to the formation of the Canfinal RG. There are also signs of lateral moraines located beneath the RG. The RG has a length of ~820 m, a maximum width of ~250 m, and a surface area of ~0.2 km$^2$. Its front is located at an elevation of ~2220 m and faces north-east. Its rooting zone begins at an elevation of ~2500 m, at the base of a ridge. This cliff links the Piz Canfinal (2812 m) to the Corno Compascio

(2808 m) and contains several weak zones feeding debris to the RG. Longitudinal ridges in the upper part are a sign of sharp contrasts in horizontal displacement between the center and the edge of the RG. The presence of longitudinal ridges in the lower section indicates a lower flow (Figure 2). The RG's lower foot is more vegetated and appears to be separated from the rest. Although it will be considered as a whole in the context of this study, the entire structure is most likely the result of several generations of RGs, some of which are now inactive or relic (Figure 2).

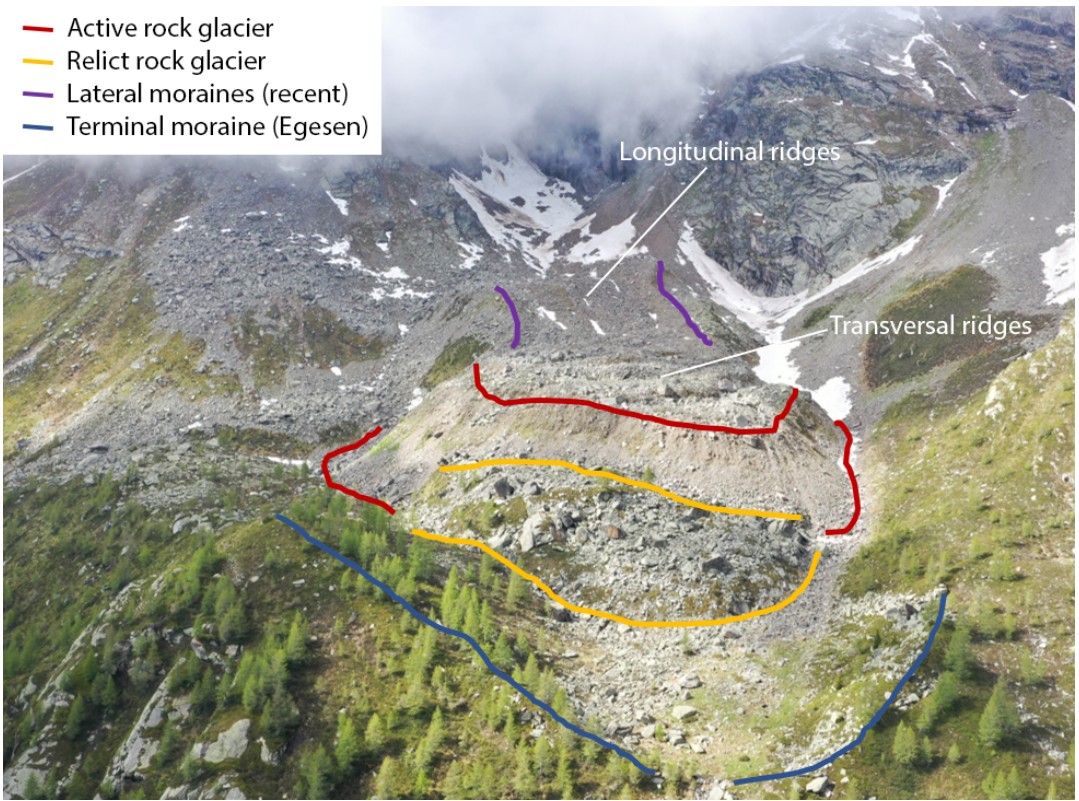

**Figure 2.** Geomorphological description of the Canfinal RG and its surroundings. Additional information on Quaternary geological and geomorphological features in the Val d'Ursé can be found in Figure S1 (SI).For reference, the front of the rock glacier is approximately 250 meters wide.

Numerous perennial and intermittent springs immediately downslope from the RG indicate a shallow and dynamic water table that reaches its maximum near the end of the snowmelt season. The spring at the foot of the Canfinal RG is located at ~2210 m in a relatively flat, humid meadow. During periods of high runoff and snowmelt, another spring is active at the mid-slope of the talus, while the majority of the springs at the foot of the talus slope are perennial and feed the Val d'Ursé.

## 3 Methodology

### 3.1 Digital image correlation

Digital image correlation is a powerful method to calculate rock glacier displacement fields (e.g., Amschwand et al., 2021) based on snow-free remote sensing imagery. We analysed the historical and present rheological dynamics of the Canfinal RG using 13 largely snow-free SWISSIMAGE ortho-rectified photos (swisstopo) from between 1956 and 2019 and data from an unmanned aereal vehicle (UAV) survey from October 2022. The ortho-photos have a pixel resolution of 0.5 m for the oldest (1956–2003), 0.25 m for the 4 images taken between 2006 and 2015 and 0.1 m for 2019. Using the *PIX4DMapper* software, the photos from our UAV survey were converted into an ortho-photo with a resolution of 2.7 cm.

We used an open-source fast Fourier transform (FFT)-based digital image correlation open source package (Bickel et al., 2018). Before co-registration, the software pre-processes images via a Wallis filter in order to reduce noise and improve contrast (Baltsavias, 1991). Displacement is calculated across image pairs. The co-registration algorithm works by selecting a reference block of pixels with sufficient optical contrast in the first (oldest) ortho-image. In the second ortho-image, the algorithm searches for the same block in a test area. If the block is successfully detected, a local horizontal displacement vector between the two images is calculated by the difference via the coordinates of the centroid. In our application, only vectors with a RMSE under a defined threshold, depending on the resolution of the orthophotos, are kept. This filtering enables exclusion of spurious data from steep slopes and in cliff-adjacent areas that are frequently covered by loose debris or snow.

### 3.2 Hydrochemistry and tracers

Spring and streams samples were sampled during seven main campaigns carried out between July 2022 and July 2024 (30.06.2022, 12.07.2022, 18.09.2022, 29.10.2022, 11.06.2023, 03.06.2024 and 11.07.2024) at different hydrological periods and when access to the site was possible. A total of 37 springs and streams were sampled at least once (Figure 3). Water samples for anion, cation and water isotopes were filtered through a 0.2 $\mu m$ cellulose acetate mesh sieve and stored in PTFE bottles for cations/anions and glass vials for water isotopes. Samples for cations were acidified. All samples were stored at $4^oC$ and analyse within a week. Cations and anions were analyzed using ion chromatography (Dionex), and water isotopes were measured with a Picarro L2130-i isotopic water analyzer at the geochemical laboratory of the University of Neuchâtel. Water isotope values are reported in ‰ relative to Standard VSMOW. The seasonal evolution of the isotopic data was compared to a local meteoric water line (LMWL) corresponding to an average of four northern regions of Italy bordering the catchment (Longinelli and Selmo, 2003; Longinelli et al., 2006):

$$\delta^2 H(‰) = 7.6 \times \delta^{18}O + 9.4 \tag{1}$$

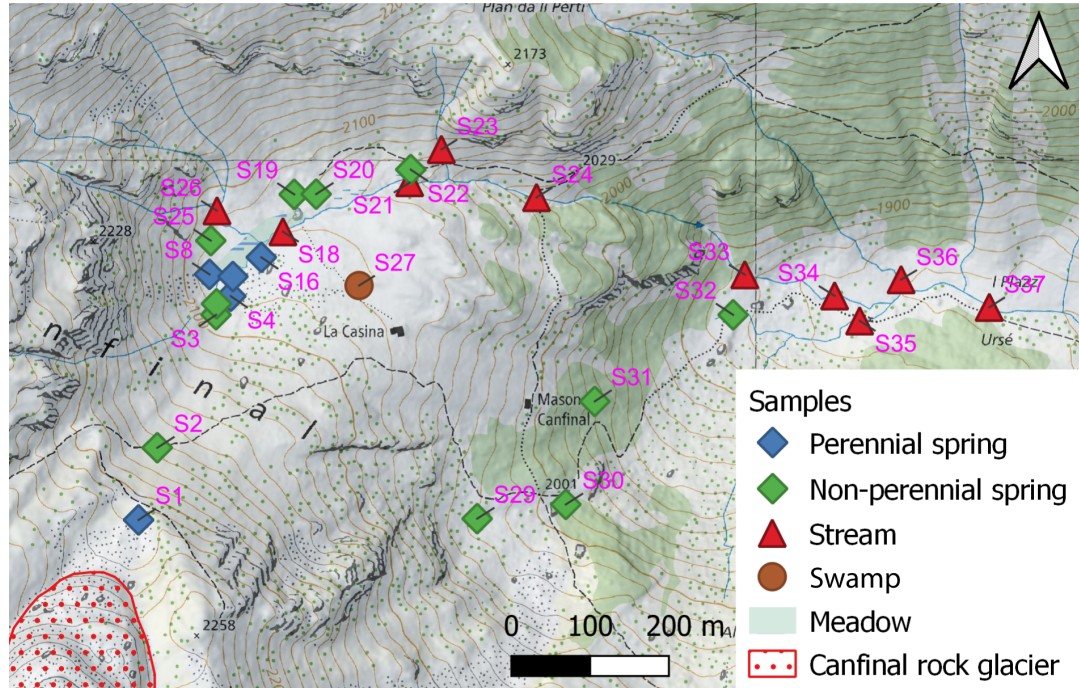

**Figure 3.** The various springs and streams from which samples were collected during five campaigns between July 2022 and June 2023 for hydrochemical and isotopic analysis. The Val d'Ursé stream gauge is at the same location as the sampling point S37. Springs S1 - S4 were also equipped with a temperature/EC probes.

Principal component analysis (PCA) was performed to statistically identify the different water sources and their varying contributions to spring flow. The variables considered included stable isotopes ($\delta^{18}$O and $\delta^2$H), EC, and concentration of the main ions: $SO_4^{2-}$, $Na^+$, $K^+$, $Ca^{2+}$ and $Mg^{2+}$. We used the *python* package *scikit-learn* (Pedregosa et al., 2011) with variable standardization (scaling and centring) for PCA of all the data.

### 3.3 Frequency-domain analyses of spring time-series

Frequency-domain analyses of hydrological time-series can offer novel insights into hydrological processes (e.g., Acworth et al., 2017; Rau et al., 2017; Houben et al., 2022). For example, they enable the direct comparison of the phase and amplitude of diurnal signals (see Figure S2). This provides information on the delay and intensity of cyclic processes. Wavelet analysis of rock glacier outflow data has been carried out by Brighenti et al. (2021), whose results hinted at the utility of such investigations. Here, we perform this analysis on data from springs that are influenced by melt water from the RG. EC from four springs situated at the foot of the Canfinal RG (S1), in the middle of the talus (S2) and at the talus/meadow interface (S3 and S4) (Figure 3) were recorded. Specific conductance (EC at 25°C) was calibrated with measurements taken by a portable probe (WTW Cond-3310). As we remove the periods during which the probes are air-exposed (i.e., dry springs) before analysis, only springs S1 and S4 have sufficient filtered data for this type of analysis. Because our frequency-domain approach utilises

relative phase and amplitudes, the absolute accuracy of the EC probes (often 5 $\mu$S/cm, which may be less than the amplitude of the diurnal variation in EC) is unimportant. Nevertheless, lower measurement noise, which may be frequency-dependent, result in more accurate determination of amplitude and phase due to higher signal-to-noise ratio.

A rolling-window FFT was used to isolate the dominant diurnal, i.e. 1 cycle-per-day (cpd) signal of the EC and air temperature ($T_{air}$) time-series. Days that were heavily influenced by dilution due to heavy rainfall events, as well as periods when the springs were dry, were excluded. A 3-day window length (Figure S2) with a 1-day step was selected to balance temporal resolution, spectral leakage, and noise reduction, as well as to avoid eliminating excessive amounts of data adjacent to excluded time-series data points.

Diurnal (i.e., frequency $f =1$ cpd) signals with non-stationary amplitude and phase dominate $T_{air}$ and $EC_{S1}$ and $EC_{S4}$ time-series. Amplitude ($A$) is a measure of the intensity of the diurnal variation, while phase ($\phi$) or, similarly, time-lag is a measure of the time of day that minimal and maxima occur. Here, to aid interpretation, we present the data as a shifted time-lag ($\Delta t$):

$$\Delta t = \frac{24\text{h}}{2\pi}\left(\phi + \frac{\pi}{2}\right) = \frac{24\text{h}}{2\pi}\phi + 6\text{h} \tag{2}$$

where $\phi$ is implicitly defined, noting the sign convention here, in the sinusoid:

$$f(t) = A\sin\left(2\pi t/(1\text{ day}) - \phi\right) \tag{3}$$

The time-lag is then limited to the range of $[0\text{h}, 24\text{h})$ by shifting values outside this range by multiples of 24 h. Thus, a sinusoid with a phase $\phi = 0$ corresponds to a time-lag of 06:00 UTC+0, i.e. the time of day at which the function is at its maximum. In a hypothetical case where maximum temperature coincides with minimum EC, the signals would be completely out of phase and thus the time-lag difference between them would be $\pm12$ hours. Interpretation of this data is discussed in Section 5.3.

## 4 Results

### 4.1 Inter-annual dynamics derived from photogrammetry

The RG horizontal velocity field (Figure 4) shows a high degree of spatial and temporal variability. For the most recent period (2019–2022), the horizontal velocity is highest in the centre of the upper part of the RG. Here the average horizontal velocity can slightly exceed 1 m year$^{-1}$. A decreased velocity is observed in the lower, central part where horizontal ridges are visible. At the edges of the RG and in the frontal zone, creep is negligible.

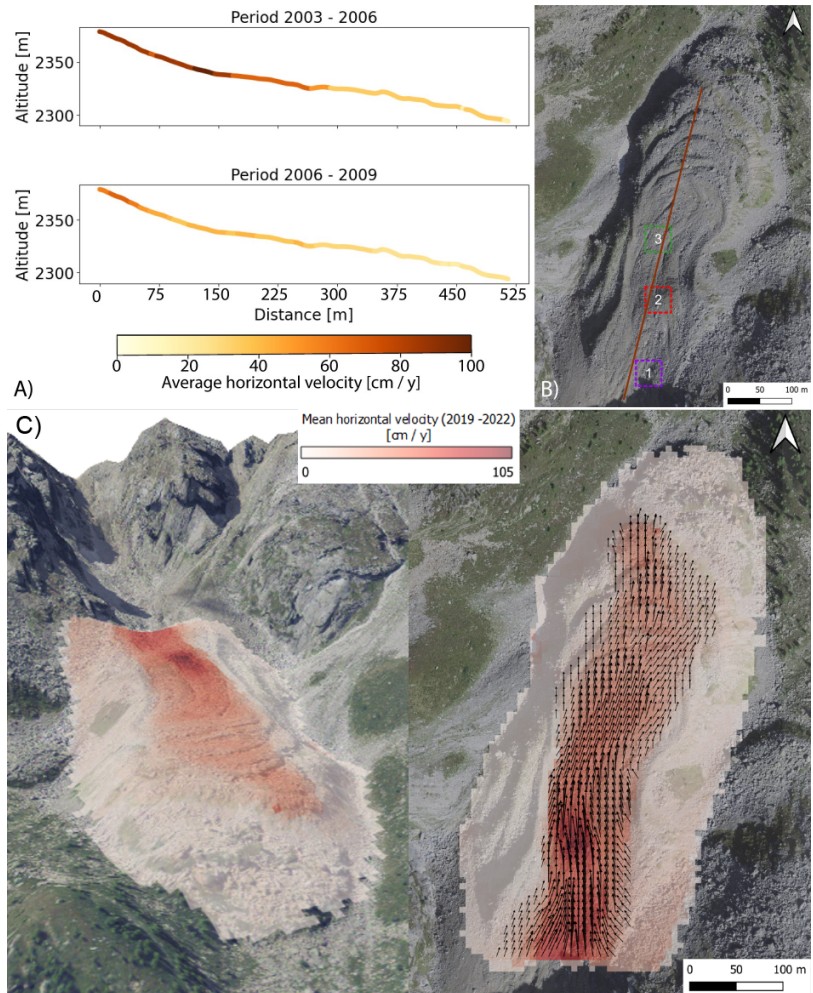

**Figure 4.** *A)* Average horizontal velocity along the topography between the contrasting periods 2003–2006 (high activity) and 2006–2009 (low activity). *B)* The three zones in the upper part of the Canfinal RG for which recent changes in average horizontal velocity have been compared (see text). *C)* 3-D and top-down views of average horizontal velocity of the Canfinal RG for the period 2019–2022. Vectors represents the horizontal displacement direction and intensity.

The differences in average horizontal velocity along the topographic profile of the RG confirm that velocities are highest in the upper part. Average horizontal velocities for three 40 m × 40 m zones in the upper part were compared with six climate indicators for the period 1990 to 2022. These climate indicators were obtained form the global scale ERA5 database (Hersbach et al., 2023), which provides data calculated on a 31 km grid. While local climate variability may be averaged out in such coarse global datasets, they still offer valuable insights into broader trends and seasonal dynamics of the required climate indicators.

– Summer temperature anomaly: Mean daily $T_{air}$ anomaly for the summer period (June–August) compared to the mean summer $T_{air}$ measured since 1940.

- 5% of hottest days: Number of summer days (JJA) per year when the mean $T_{air}$ is higher than 95% of the daily summer mean $T_{air}$ measured since 1940.

- Winter days: Number of days per year when the average $T_{air}$ is below 0°C.

- End of 10 cm snow cover: date when the snow cover falls below 10 cm.

- December snow depth anomaly: anomaly of the mean snow depth at the beginning of the winter season, i.e. in December.

- 5% of wettest days: Number of days per year on which the cumulative precipitation is greater than 95% of the daily cumulative precipitation since 1990.

The results show a general acceleration of the average horizontal velocity since the 1990s (Figure 5), especially for zones 2 and 3 (Figure 4b). The average horizontal velocity in zone 2 was $39\pm6$ cm year$^{-1}$ between 1992 and 2003, while its maximum was measured between 2019 and 2022 with $71\pm5$ cm year$^{-1}$. In zone 3, the average velocity was $32\pm6$ and $52\pm5$ cm year$^{-1}$ for the same two periods, but peaked at $72\pm9$ cm year$^{-1}$ between 2003 and 2006. The variations are less significant for zone 1, which nevertheless experienced a slowdown in the period 2010 - 2012 ($37\pm12$ cm year$^{-1}$). This stabilisation is also visible in zone 2.

Only the climate indicators for the summer $T_{air}$ anomaly and days with extreme temperatures show a similar overall trend that can be attributed to climate change. All three zones experienced a strong acceleration between 2003 and 2006, with 2003 being characterised by a very warm summer $T_{air}$ anomaly ($>2$°C) and a high number of days (24) with extreme temperatures. These values correspond to the European heat wave of 2003. It should also be noted that 2014 had a strong winter with a significant snow depth anomaly that lasted until May. There were also less intense precipitation events, and the summer was characterised by a negative $T_{air}$ anomaly and few extremely warm days. In the following years, from 2015 to 2019, there was a slowdown in zones 1 and 3 and a stabilisation in zone 2. Otherwise, there are no globally similar trends between the acceleration and stabilisation phases of the three zones and the winter indicators or with intense precipitation events.

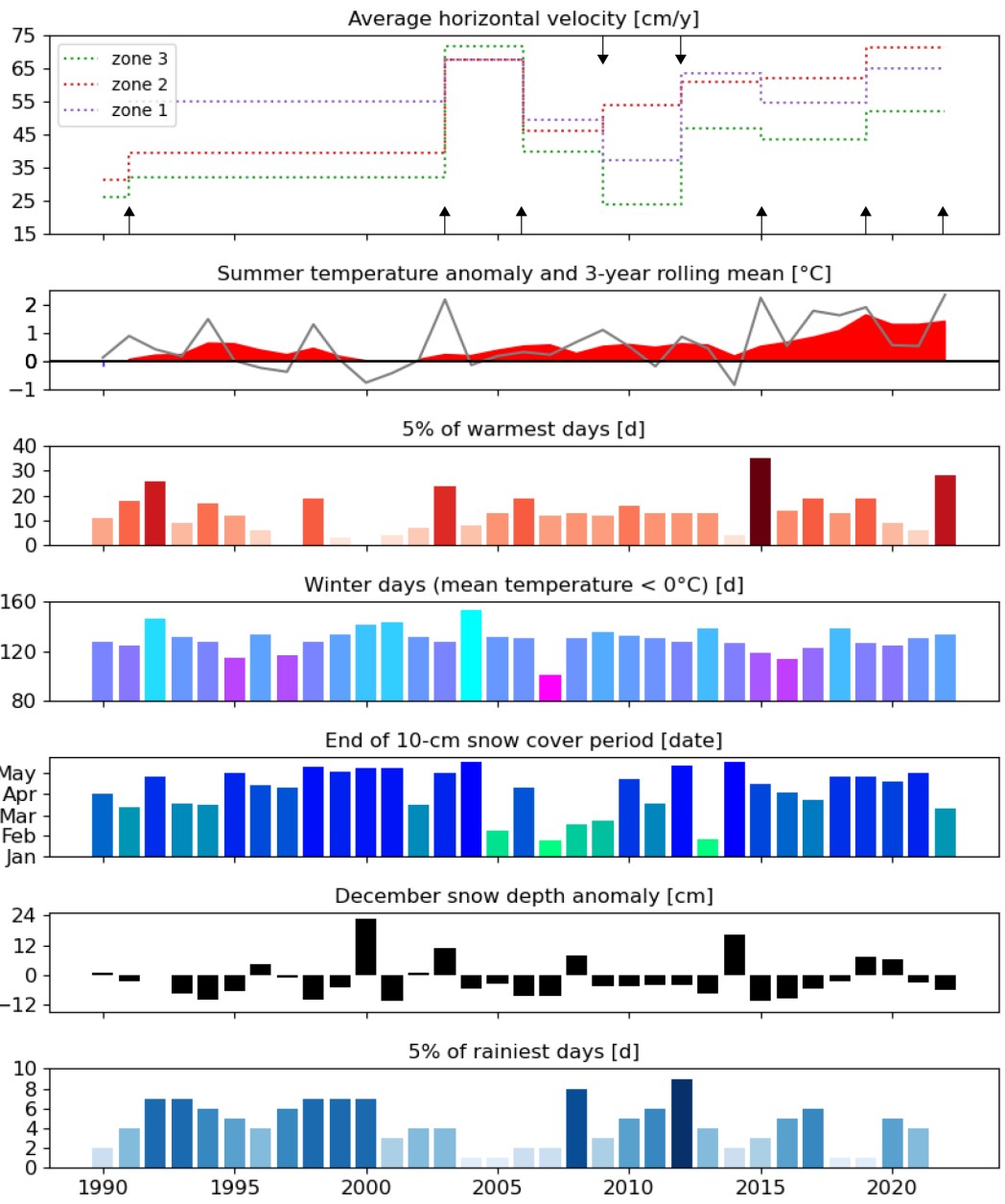

**Figure 5.** Recent evolution (1990–2022) of the Canfinal RG zonal dynamics in comparison with various summer and winter climatic indices. The 7 orthoimages available since 1990 and the drone acquisition in 2022 are indicated by an arrow. For zone definitions, see Figure 4b.

## 4.2 Spring water sources and seasonal dynamics

All ionic concentrations are relatively low for the Val d'Ursé catchment (EC $<250\ \mu S/cm$). This is expected for such alpine environment dominated by crystalline bedrock aquifers. Here, we focus on springs influenced by the rock glacier, distinguishing between those located near its front (S1 and S2, shown as triangles in Figure 3) and those at the base of the talus (S3 and S4, shown as squares in Figure 3). Additionally, we differentiate samples collected during high-flow periods (defined from May to August, indicated in blue in Figure 3) from those collected during low-flow periods (defined from September to December, indicated in red in Figure 3).

Firstly, these 2 groups of springs show clear differences in mineralisation content. S1-S2 are less mineralized than S3-S4, as evident for $Ca^{2+}$ + $Mg^{2+}$ and $SO_4^{2-}$ concentrations in Figure 6A). These differences suggest more diluted groundwater close to the RG (referred as *rock glacier end-member*) while mineralisation increases with distance from the RG, suggesting greater contribution from deep mineralized groundwater (referred as *deep groundwater end-member* in the following). However, S1 and S2 show higher concentrations of some reactive elements like $K^+$ and $NO_3^-$ (Figure 6B). S1 has the highest concentration of $NO_3^-$ ($\sim$0.03 meq L$^{-1}$) and it decreases along the talus/meadow complex as a function of distance from the RG. S4 has a $NO_3^-$ concentration of 0.022 meq L$^{-1}$, while all other springs located in the meadow area, but not aligned with the RG and the talus, have $NO_3^-$ concentrations ranging between 0.002 to 0.133 mEq.L$^{-1}$.

In a classical bivariate plot of water isotops showing $\delta^{18}O$ and $\delta^2H$, all samples align with the local meteoric water (LMWL) (Longinelli and Selmo, 2003; Longinelli et al., 2006). There is not clear distinction in isotopic compositions between the two end-members suggesting a similar source of recharge water. The only variability arises during changes in seasonal flow regimes.

Indeed, both end-members show dilution/enrichment dynamics throughout the season. During high flow periods, after the onset of the snow-melt, both end-members display a drop in concentration due to mixing with melted water. Focussing on the relationship between $Ca^{2+}$ + $Mg^{2+}$ and $SO_4^{2-}$, the deep groundwater end-member (squares in Figure 6), show a clear mixing toward the rock glacier end-member at high flow regimes of about 40%. Similarly, water isotops show an decrease in $\delta^{18}O$ and $\delta^2H$ after the snowmelt typical of cold water recharge (Figure 6C). As the season advance, the isotops show a progressive increase in $\delta^{18}O$ and $\delta^2H$.

All those observations are effectively summarized in a Principal Component Analysis presented in Figure 6D. The main elements discussed before are considered, i.e. EC, $Ca^{2+}$, $Mg^{2+}$, $SO_4^{2-}$, $K^+$, $Na^+$, $\delta^{18}O$ and $\delta^2H$. The first two axes of the PCA explain 53% and 29% of the total variation in the data set respectively. The PCA confirms the differenciation of the 2 end-members along the PC1 dominated by EC, $Ca^{2+}$, $Mg^{2+}$, $SO_4^{2-}$, while PC2 influenced by $K^+$, $Na^+$, $\delta^{18}O$ and $\delta^2H$ explain the seasonal variability with increasing values during low flow regimes.

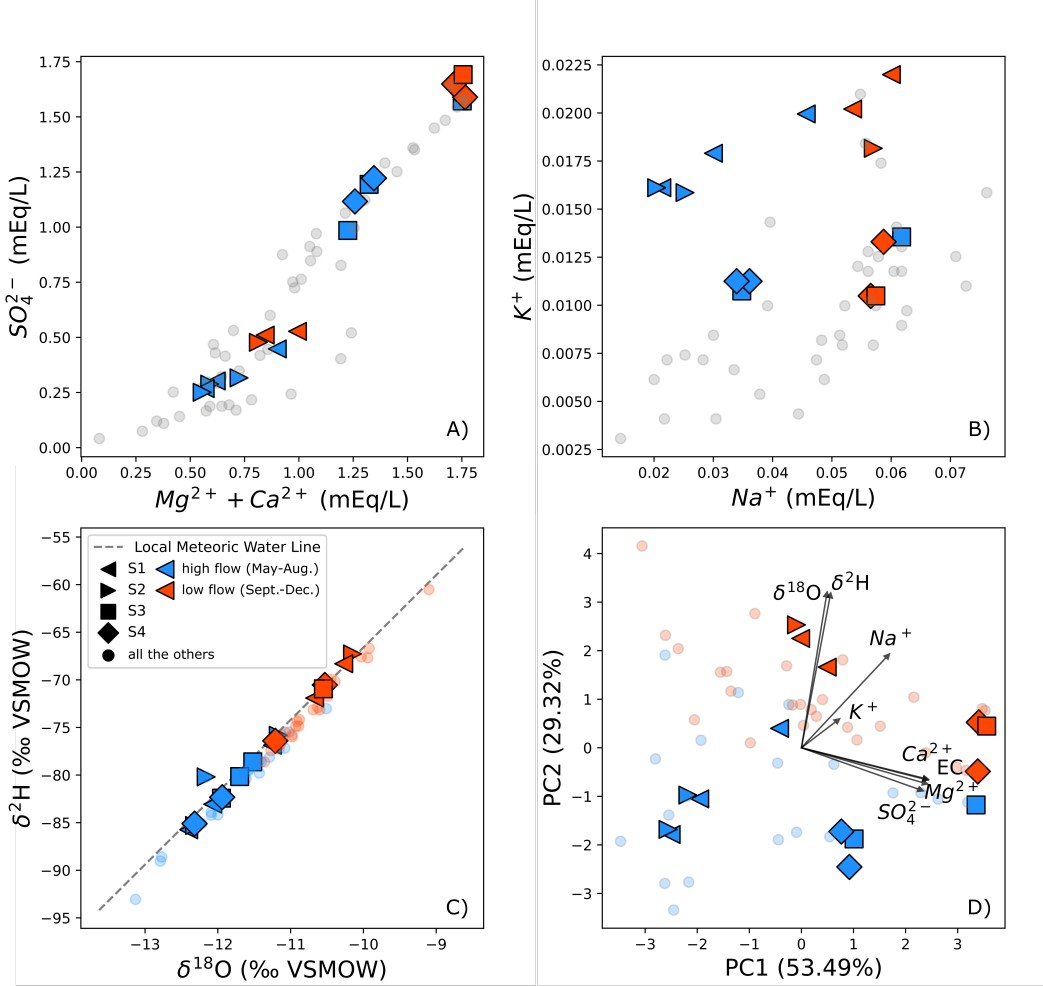

**Figure 6.** Chemical composition and isotopic ratios of main springs (S1–S4) under high and low flow regimes. (A) Bivariate plot of Sulfate versus the sum of Calcium and Magnesium concentrations. (B) Bivariate plot of Potassium versus Sodium concentrations. (C) Isotopic composition compared with the Local Meteoric Water Line (LMWL) (Longinelli and Selmo, 2003; Longinelli et al., 2006). (D) Principal Component Analysis (PCA) plot with the first two components, which explain the majority of the variance within the dataset (percentage of variance explained is shown in brackets). Eigenvectors representing the influence of each variable are indicated by black arrows. For all plots, main springs S1 to S4 are highlighted with unique symbols, while all the other sampling points are represented by circles in the background. Analyses performed during high flow regimes are marked in blue, and those from low flow regimes in red.

## 4.3 Frequency-domain analyses of dilution linked to RG diurnal thaw cycles

EC time-series measurements from the springs (Figure 7) showed varying responses at the foot of the RG (S1) and at the talus/meadow interface (S4). Rainfall events have a strong dilution effect on both springs, specifically at the end of summer and during fall. The EC systematically systematically between two events recovers to its value befor the event with a characteristic

timescale faster for S1 than S4. A seasonal variability in the EC signal is also involved at both springs, although more evident in S4. A decrease in EC is involved at the onset of the snowmelt, with minimum values around May–June, and followed by

an exponential increase until the next snow-melt recharge season. In fall, some long duration of free rain events implied that the spring S1 dried-out between September and October months. S1 is also dry during most of the snow cover period, while S4 continues to flow. In 2023, S1 was reactivated at the beginning of May following the snow-melt recharge, with a lower EC (about 40 $\mu$S cm$^{-1}$) than before the winter.

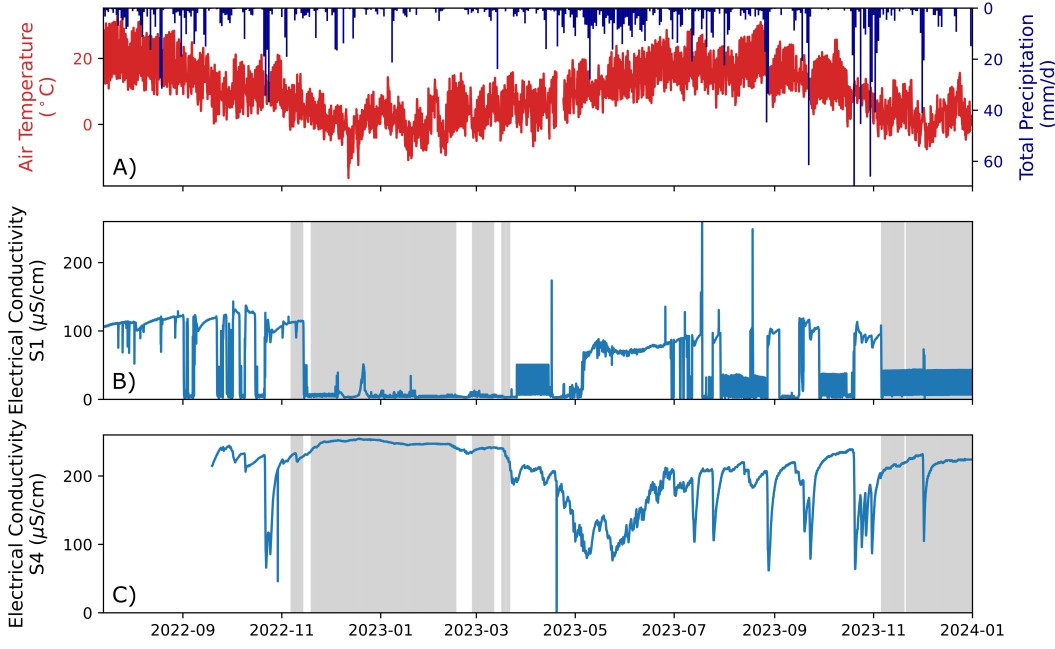

**Figure 7.** (A) Daily precipitation and air temperature from the nearby weather station Robbia (MeteoSwiss). (B) Electrical conductivity of spring S1, located at the base of the Canfinal rock glacier, and (C) electrical conductivity of spring S4, situated at the talus/meadow interface. Data are displayed from July 2022 to January 2024. Gray shaded areas denote periods when the weekly average air temperature is below 0°C. Starting April 2023, S1 EC exhibited significant noise probably linked to the sensor not being immersed in water.

The amplitude (Figure 8A-E1) and time-lag (Figure 8A-E2) of the 1 cpd components of $T_{air}$, $EC_{S1}$ and $EC_{S4}$ were

calculated using a 3-day rolling window stepped in increments of 1 day. Frequency-domain analysis of these time series reveals significant amplitude variations and a phase lag between the $T_{air}$ and EC signals measured at these springs. The 1 cpd amplitudes vary significantly for all datasets, indicating seasonal differences in the intensities of daily cycles. $T_{air}$ varies more diurnally during summer than winter, as indicated by its amplitude trend. The amplitudes of the spring EC diurnal variations also exhibit seasonal trends, which may be linked the intensity of dilution from RG melt. S1 and S4 show both a much higher

amplitude in period immediately after snowmelt than it does in the summer and autumn period, with an amplitude of 0.75 for S1 and 2.25 for S4 in June. Then amplitudes decrease to values close to 0 in the late summer and fall. For S1, this low amplitudes were particularly visible in 2022 while 2023 had poor quality data which prevented the analysis. For S4 this seasonality is

consistent during the full recording period (see Section 5.4). This seasonality is also assessed by looking at the ratio of the EC amplitudes to those of $T_{air}$, $EC/T_{air}$ which aims at normalising the EC amplitudes by the main driver of daily melt rate variations (Figure 8D-E1). Note that S4 shows significant variations in late summer, due to the high variability in EC amplitudes at this period, probably due to high precipitation storm events influencing the signal.

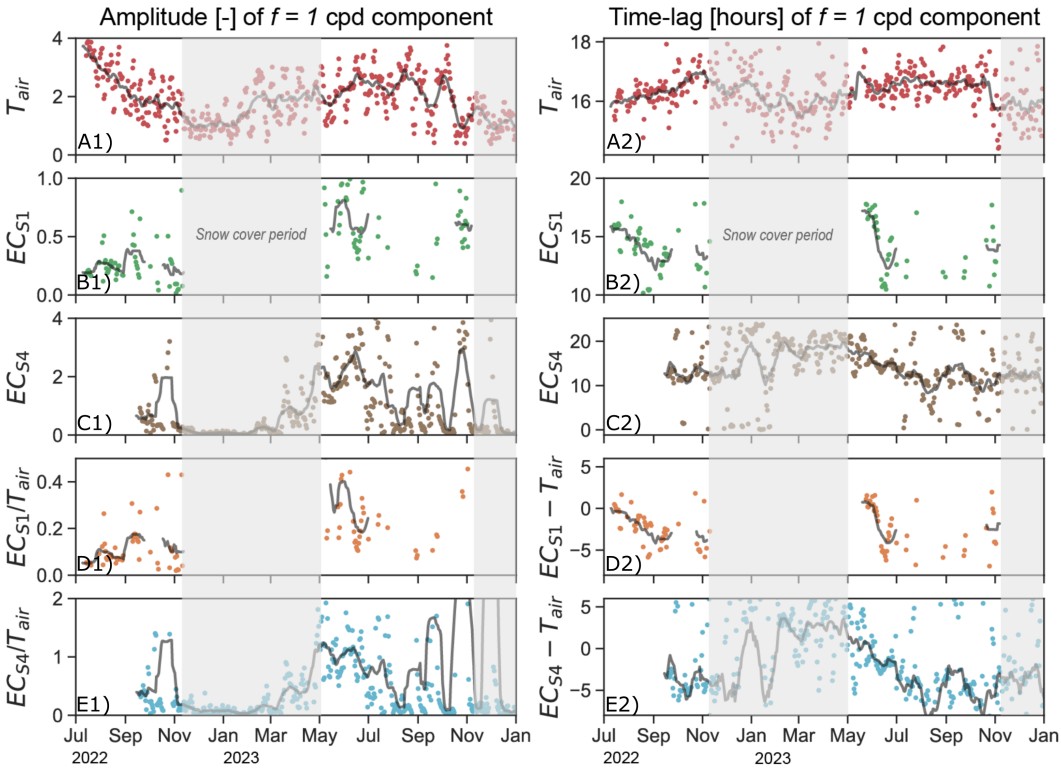

**Figure 8.** Amplitude (A-E1) and time-lag (A-E2) of 1 cpd components of the filtered EC time-series from springs S1 (foot of the Canfinal RG) and S4 (talus/meadow interface) and air temperature between July 2022 to January 2024. The relative amplitude ratios (D1-E1) and time-lag differences (D2-E2) between the springs EC and air temperature signals are also illustrated. Rolling means with a 21-day window length are shown as continuous line in order to help discern seasonal variations. Here, amplitudes can be considered unitless and phase has been converted to time-lag (i.e., from radians to hours UTC+0) to facilitate interpretation (Section 3.3). The periods of snow cover are indicated by the shaded sections.

The time-lag information (Figure 8A-E2) enables an objective measurement of the timing of diurnal dilution processes. The phase of the 1 cpd component of $T_{air}$ is, as expected, relatively stable throughout the year. S1 exhibits a clear decreasing trends in time-lag (Figure 8B2) starting at the onset of snowmelt, from ∼15h till ∼12h indicating that the times of minimum daily EC occur progressively earlier over this period. The phase of S4 shows a similar trend over the same period from ∼19h till ∼11h, particularly visible in the 2023 spring. The time-lag difference (or phase shift) between EC and $T_{air}$ (Figure 8D-E2) confirms

these observations. Nonetheless, we can observe that the EC daily maxima generally precede the $T_{air}$ maxima and thus the EC daily minima lag $T_{air}$ by <12 hours.

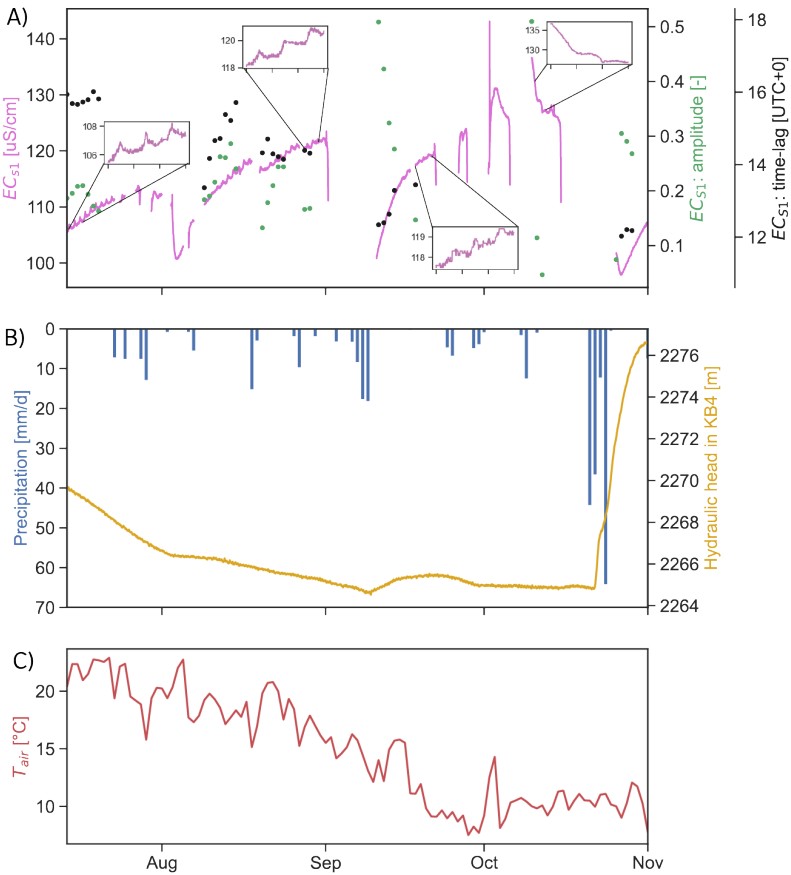

**Figure 9.** A) Cleaned electrical conductivity of spring S1 as well as the amplitude and time-lag of the 1 cpd frequency component (see Figure 8) for the period 14.07.2022-01.11.2022. Periods with no data represent days with precipitation or spring flow cessation. Detail of selected 3-day windows is also shown to illustrate the variations in daily variations. B) Fractured aquifer hydraulic head (KB4, see Figure 1) and daily rainfall (mm) over the same time period. C) 7-day average air temperature (° C.

The longer-term, non-diurnal variations in EC and hydraulic head provide valuable insights into local hydrological processes
(Figures 9 and S4). During the summer and autumn snow-free period, the EC of springs S1 and S4 gradually increases (Figure 9A), while the hydraulic head measured in borehole KB4, adjacent to the rock glacier (Figure 1), decreases. The minimum hydraulic head, recorded in early September and late October at KB4, corresponds to a period when spring S1 was dry, while S4 continued to flow throughout the season. The overall cooling trend in air temperature during this period is associated with a decrease in EC amplitude and a negative shift in time-lag at S1, highlighting that lower temperatures lead to a reduced rate of
rock glacier thaw. Precipitation events further influence temporally the seasonal variability by recharging the deep aquifer and

diluting the spring EC signal. These punctual events impact the frequency-domain analysis, modifying both the amplitudes and time-lags of EC variations before returning to previous values.

## 5 Discussion

### 5.1 Creep is accelerating as a result of rising temperatures

Horizontal velocity patterns of Canfinal RG are consistent with the presence of longitudinal ridges in the upper part of the RG, indicating strong velocity contrasts between the centre of the RG (1 m year$^{-1}$) and the edges. The horizontal ridges in the lower part confirm the stabilisation of the displacement. We can conclude that the upper part is active, while the lower part consists of older features that are now inactive or relics. Since 1990, the zones delineated in the upper active part, especially zones 2 and 3 (Figure 4b), which are further away from the cliffs, have accelerated. This is the steepest part of the RG and also the

zone for which most conceptual models indicate the largest proportion of ice (Jones et al., 2019; Schaffer et al., 2019), making this higher velocity a good indicator of the presence of a large amount of supersaturated debris/ice mix under steady-state creep. This demonstrates that, in addition to slope, the volume and temperature of the ice, which affect its rheology, as well as the increase in liquid water content caused by thawing, have significant impact on the dynamics of RGs (Duval et al., 1983; Kääb et al., 2007; Marcer et al., 2021). Nonetheless, most of the horizontal RG displacement is observed in the shear horizon.

It is located at the base of the RGs and the effect of $T_{air}$ is therefore delayed and limited (Kääb et al., 2007; Cicoira et al., 2021). Temperature variations within the shear horizon are thought to alter its mechanical properties over longer time scales, particularly those near the melting point. The presence of pressurised water within this unit suggests that the unfrozen water content has a strong influence on its dynamics (Ikeda et al., 2008; Buchli et al., 2018; Cicoira et al., 2021). In addition, melting ice allows more liquid water to enter the shear horizon, with the associated destabilising effect and heat conduction (Wirz et al.,

2016).

Although the determined horizontal velocities are multi-year averages, the trends appear correlated to the summer $T_{air}$ anomaly and the number of days with extreme $T_{air}$ (Figure 5), both of which are increasing in the Alps due to global warming. This observation is consistent with the results of several studies in the Alps indicating acceleration of RGs as a result of rising temperatures (Kääb et al., 2007; Marcer et al., 2021). We note, however, that recent deceleration has been observed at four

RGs, at similar elevations to the Canfinal RG, but situated immediately down-gradient of glaciers, in the Swiss National Park (Manchado et al., 2024). 2014 saw a colder summer, and it appears that this may have had a stabilising effect in the following years. In fact, this phenomenon was also noted by Cusicanqui et al. (2021) for the Laurichard RG, which is underlain by highly fractured granite (Bodin et al.). A strong acceleration after the European heat wave of 2003 was also observed for many other RGs in the Swiss Alps and western Austria (Krainer and He, 2006; Delaloye et al., 2010). These similarities suggest that active

RGs in the same geographical area tend to be affected in the same way by summer temperature extremes. Although it has been shown that precipitation can cause short-term acceleration of RGs (Wirz et al., 2016), our results lack the temporal resolution to confirm this. There is no visible relationship between the average velocity changes and the winter indicators. One explanation could be that the long-term activity of the RG is mainly influenced by $T_{air}$ and/or that the other parameters have no significant

effect in comparison. It is also possible that confirmation of stabilising or accelerating effects of the various indicators (Figure 5) requires data with a higher temporal resolution.

In addition, the data available at the Canfinal RG do not allow for establishing a clear relationship between multi-year kinematics and climatic or hydrological conditions, although links between hydrology and kinematics evident. As suggested by Cicoira et al. (2019), recharge from precipitation and snowmelt is one of the main drivers for RG displacement. The investigation of these coupled processes offers interesting prospect for future research, especially as global warming increasingly influences precipitation patterns, including shifts in the balance between rain and snow in the Alps. Observations of increased displacement in areas where the morphology indicates ice-rich permafrost suggest that these zones could follow the same trajectory as the glacier's front under the effects of warming: a gradual reduction in ice content until becoming immobile.

## 5.2 Seasonal variations in spring end-members reveal freeze-thaw cycle and transport dynamics

A progressive enrichment in heavy stable isotopes is observed for all springs and streams sampled as the summer advance, while the hydraulic head of the fractured aquifer and the streamflow measured at the Val d'Ursé decrease during the same period. This indicates that the contribution of groundwater increases after the end of snowmelt and maintains the baseflow of the catchment. This enrichment can also be explained by a higher contribution of enriched meteoric water formed at higher temperatures than winter snow, increased evaporation from open surfaces in summer and/or the release of meltwater from permafrost that has undergone several freeze/thaw cycles and induces recharge at higher altitudes (Williams et al., 2006; Moran et al., 2007; Beria et al., 2018; Brighenti et al., 2021). The seasonal variations in EC measured at S1-S4 as well as in the outlet streamflow of the Val d'Ursé catchment confirms that the water of the fractured aquifer is more mineralised during the winter low flows while it is significantly diluted from snowmelt recharge, rainfall events and permafrost thawing from spring to fall (Colombo et al., 2019). An clear contribution from the Canfinal RG cannot be directly detected in the signals from borehole KB4 nor in the EC or streamflow variations measured at the outlet of the catchment, and thus at the headwater catchment scale. This is in agreement other RG studies that have determined contribution to total discharge by active RGs to be $\lesssim 5\%$ (Krainer et al., 2015; Harrington et al., 2018; Bearzot et al., 2023).

The PCA performed on a single sampling capmpaign performed in October 2022 samples (Figure S3) shows that the springs closest to the Canfinal RG have a distinct geochemical signature from the others. They are characterised by a higher concentration of $K^+$ and $Na^{2+}$ and the greater enrichment of heavy stable isotopes. The fact that rock weathering is thought to be the main driver for the presence of certain trace elements in RG runoff helps to explain the high concentration of $K^+$ and $Na^{2+}$ (Colombo et al., 2018). In fact, the rocks that make up the debris of the RG and the surrounding cliffs are orthogneiss (Sella granodiorite) with muscovite and feldspars, and a granite intrusion (Musella granite) rich in biotite and feldspars. The concentrations of $NO_3^-$ are higher between the RG and the talus-meadow interface than in other springs in the Val d'Ursé catchment. Studies have highlighted this peculiarity in the discharge of RGs and blockfields (Williams et al., 2007; Colombo et al., 2019). Some suggest that bacterial activity in RGs is the cause of this enrichment (Fegel et al., 2016; Tolotti et al., 2020). These extreme environments, with snow cover in winter, low light levels and stable, cold soil temperatures around $0°C$, are carbon limited and can support nitrifying bacterial communities. According to Del Siro et al. (2023), it could also be atmospheric

pollutants released by melting ice. These results lead us to conclude that the presence of the Canfinal RG and the lithology of its constituent rocks influence the hydrochemistry of the springs fed by its discharge. The higher isotopic values of the discharge from the RG could be explained by easier infiltration of water from summer rainfall events into it (Krainer et al., 2007), or by an altitudinal gradient, as proposed by Moran et al. (2007).

### 5.3 RG diurnal dynamics are revealed through frequency-domain analysis

The EC measurements of springs S1 and S4 during the snow-free period shows the existence of a daily cycle of dilution/enrichment. Dry periods are characterised by an increase in the EC of the discharge over several days, influenced each day by a dilution that starts in the afternoon. Although the discharge rate could not be measured, available studies indicate a maximum discharge rate in the afternoon, corresponding to the meltwater supply of the RG (Krainer and Mostler, 2002). We note that both low-EC (e.g., Haeberli, 1990) and high-EC (e.g., Nickus et al., 2023) ice has been observed in RGs. Furthermore, EC is not a conservative tracer and contrasting mechanisms, often related to short-term weather events, of RG EC-discharge relationships exist (Colombo et al., 2018). As we filter out spikes in EC caused by rain events, our hypothesis is that our observed EC variations are related to diurnal melt cycles. Due to the absence of high-solubility rocks in the rooting zone of the Canfinal RG, we make our interpretations based on the hypothesis of lower-EC discharge. The majority of melt in the snow-free period is assumed to originate from the active layer, but we recognize that in the late summer and autumn, it is possible that significant melt from "older" permafrost may occur (Del Siro et al., 2023).

Under these assumptions, the diurnal RG discharge peak is expected to correspond to the S1 EC minimum due to its dilution effect. This occurs with a time-lag difference (or phase-shift) relative to the maximum $T_{air}$ due to *a)* the residence times of the flow path and *b)* the heat transfer and melt dynamics in the RG. We observe a clear trend in the lag between the S1-S4 EC and $T_{air}$ diurnal cycles during the snow-free period (Figure 8). The shift from $\sim$0 hours in July to $\sim -5$ hours for S1 in November implies a lag between maximum $T_{air}$ and minimum EC of $\sim$12 hours during the warmest period and $\sim$7 hours during the coldest snow-free period. In 2023, there is a clear trend in the S1 time-lag difference relative to $T_{air}$ (lower right, Figure 8) from $\sim$+1 to $\sim$-5 hours over the May-August period, implying a lag between maximum air temperature and minimum EC of $\sim$13 hours at the beginning of the snow-free period that decreases to $\sim$7 hours by mid-Summer. After this, the time-lag fluctuates with no clear trend.

There are likely several factors that cause these seasonal trends in lag time between diurnal $T_{air}$ maxima and EC minima. Firstly, the active layer thickens over the snow-free period, implying increasingly shorter flowpaths throughout the melt season. Furthermore, preferential flow paths, akin to moulins in glaciers, may also be formed internally by melt water throughout this period, further reducing the residence time. This particular phenomenon, which we cannot confirm with our current data, would also have the potential to initiate and accelerate melt from older from permafrost which could have a higher EC than the active layer. Secondly, as temperatures decrease, there will be, proportionally, less melt from higher-elevation zones of the rock glacier and thus the mean flow path length to the foot of the RG will decrease. Finally, during the warmest months, melt is expected to be continuous, while in the late season daily minimum temperatures are often $<0°$C. We posit that these processes combine to contribute to a delay between maximum air temperature and minimum spring EC that decreases throughout the

snow-free period. For S4, while the spring/summer trend in time-lag is clear, there is the additional complexity of likely significant influence from other sources. The proportion of deeper groundwater likely increases throughout the first half of the snow-free period, while the water originating from RG melt has a decreased effect. The S4 EC data, which show a decreasing diurnal amplitude over the snow-free period, support this hypothesis. As the Canfinal RG was previously undocumented and unexplored, there are unfortunately no data available on the RG internal structure and ice chemistry. Future electrical resistivity tomography (e.g., Mewes et al.; Buckel et al., 2023) and core drilling investigations could provide data to support our above-outlined reasoning.

## 5.4 Intermittent flow and dilution intensities in springs

In the southern European Alps, the summer and autumn of 2022 were drier than average and followed a winter with exceptionally low snow accumulation (Choler, 2023) with again a dry 2023 summer. Because of this, the contribution of deeper groundwater to the springs (Figure 10) may have played a more important role than in average years. As discharged baseflow becomes more mineralised, the amplitude of dilution due to the diurnal freeze-thaw cycle of the RG ice may increase. At the end of winter, S1 reactivates as the snowpack melts, resulting in a large diurnal amplitude that is also visible in the spring discharge at the talus/meadow interface. This emphasises the importance of both snow- and ice-melt cycles as the cause of important oscillations in water parameters (Krainer and Mostler, 2002; Berger et al., 2004; Krainer et al., 2007) and the fact that these cycles smooth out during the summer (Brighenti et al., 2021). Based on these results, groundwater from the shallow fractured aquifer is the main component of the Canfinal RG discharge (Figure 10). Its discharge is strongly influenced in springs and until summer by snowmelt, which reactivates S1 with a daily dilution of greater amplitude and occurring earlier in the afternoon. Intermittent drying of springs S1–S3 was observed multiple times, with springs S2 and S3 (in the talus slope) only flowing continuously at the beginning of the snow-free period. These observations combined with the pore pressure data monitored at the KB4 borehole (Figures 9 and S4) indicate that groundwater levels are only high enough to sustain spring discharge immediately after snowmelt before the water table drops. Flow in springs S2 and S3 later in the year is short-lived and mainly caused by precipitation events. In contrast, spring S1, as discussed, is directly influenced by RG thawing which contributes to a groundwater table that remains at or near the surface for a longer period. Spring S4 is at the talus-meadow interface where the topographic gradient lessens and, as such, appears to have perennial flow assured by the deeper groundwater system even when the water table is at its annual minimum. This water has a longer residence time, as indicated by its higher mineralisation compared to the other three springs. During winter baseflow conditions, snow cover and large amounts of frozen ground prevent recharge of the fractured aquifer. As a result, the water table drops significantly and all springs are dry except for S4 (Figure 10).

## 5.5 Refining the conceptual model for RG-groundwater interactions

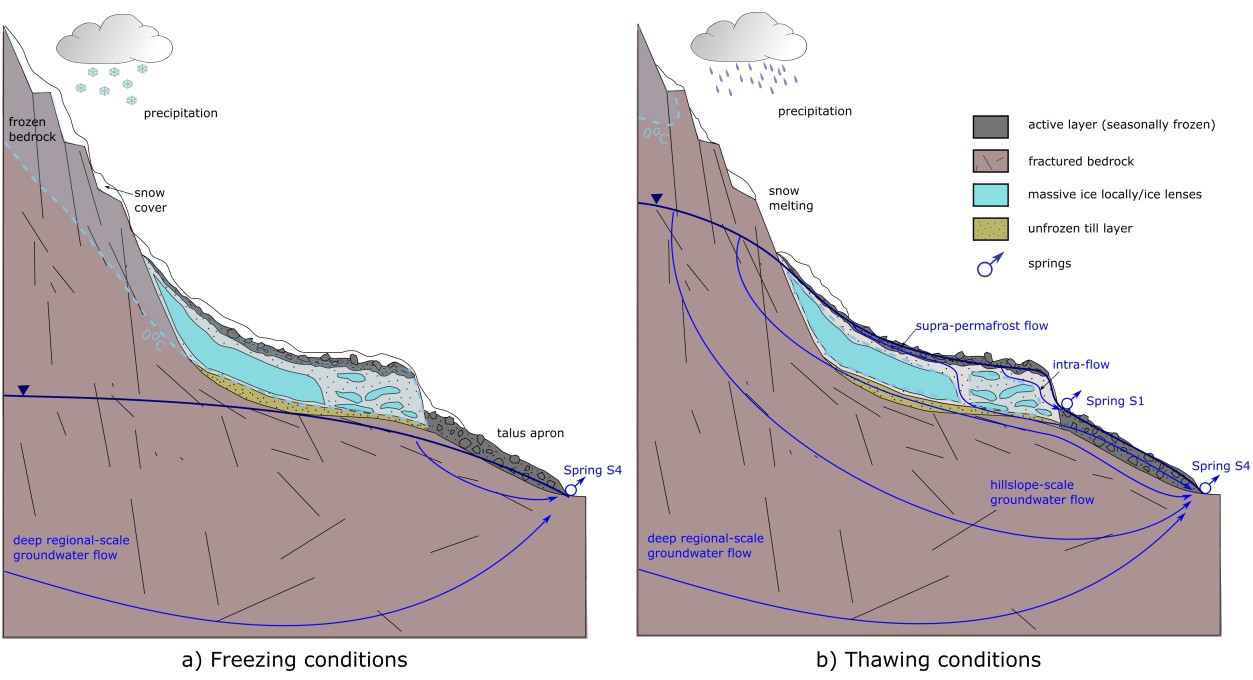

**Figure 10.** Conceptual model of the interaction between permafrost and mobile groundwater at the Canfinal RG during a) freezing conditions in winter and b) thawing conditions in summer. Modified from Jones et al. (2019). The figure is not to scale.

The ensemble of our analyses allows us to propose a refined conceptual model for the cryo-hydrogeological functioning of the Canfinal RG (Figure 10). Freeze/thaw cycles influence spring runoff and solute transport on seasonal and daily time scales. During winter, the shallow subsurface remains dominantly frozen, preventing groundwater recharge and confining groundwater flow to deep flow paths with longer residence times and higher solute concentrations. As temperatures rise during the summer, the active layer of permafrost thaws, re-enabling shallow groundwater flow paths. Meltwater from ice-rich RGs and permafrost discharges to down-gradient aquifers, springs and streams. Spring discharge rates and their composition in specific environmental tracers such as EC, ions and water stable isotopes vary as a result of recharge and freeze-thaw cycles. Melting of low EC ice from the RG results in dilution of down-gradient springs with effective residence times progressively decreasing until the return of freezing conditions with low water table.

## 6 Conclusions

Although RGs generally contain smaller quantities of water than glaciers (Wagner et al., 2021), their hydrological role is significant in many alpine catchments. Their rheological and hydrological state serves as an important climate change "barometer" for alpine environments where they are present. As climate change is expected to increasingly impact mountain headwater

catchments (van Tiel et al.; Evans et al., 2018; Somers et al., 2019; Arnoux et al., 2021; Halloran et al., 2023), it is urgent to improve our understanding of the feedback mechanisms between the cryosphere and hydrology, where the interface between permafrost and groundwater is critical. We confirmed that the Canfinal RG is active and that its creep rate appears to have accelerated in recent years alongside increasing temperature as comomnly observed in the Alps and other mountain regions worldwide. Without measurements of the internal structure of the RG, it is not possible to reliably estimate its remaining "lifespan". Geochemical and isotopic analyses enabled the differentiation of water sources contributing to the connected springs. The frequency-domain analyses in which we isolate the non-stationary 1 cpd components of EC and temperature provided valuable insights into the connectivity of the RG and springs. We have determined that the delay between diurnal air temperature maxima and dilution minima decreases after the snow-melt recharge, implying a progressive decrease in RG-spring residence times, and have proposed several mechanisms for this phenomenon. We recommend that this type of approach be applied to additional sites to tests its limits in determining the temporal dynamics of permafrost-groundwater interactions. Monitoring of the RG interior via direct measurements or geophysical techniques, as well as numerical modelling, may provide us with a refined understanding of the internal hydrological processes in the Canfinal RG and other RGs. Our combination of remote sensing imagery, geochemical measurements, and hydrological monitoring has enabled a more comprehensive understanding of this vulnerable alpine cryo-hydrological system.

*Author contributions.* All authors wrote the manuscript collaboratively. CL: field work, laboratory analyses, data validation and analyses, figure creation, programming, and synthesis. LJSH & CR: supervision, field work, data analysis, programming, figure creation, synthesis, project management, and revisions. CR: Data management and coordination activities of the Val d'Ursé Critical Zone Observatory.

*Competing interests.* The authors declare no competing interests.

*Acknowledgements.* We thank handling editor, Geneviève Ali; 2 anonymous reviewers; and community commenter, Giacomo Medici, for their feedback which has improved the final manuscript. We would like to thank Luc Illien (GFZ Potsdam), Emilio Kenda, Noam Makkinga et Eliot Barbier (Université de Neuchâtel) for their assistance in the field, as well as John Molson (Université de Laval) for the discussions about thermal-cryosphere processes. We are grateful to Clémence Berguerand and Isaline Perrenoud for perfoming the geochemical analyis at the CHYN laboratory. CR and LJSH acknowledge funding from the Waterwise project, co-funded by the European Union through the Interreg Alpine Space programme. LJSH acknowledges funding from the Swiss National Science Foundation (SNSF Grant #212622, "RADMOGG").

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
