# Peer review of "Seasonal and diurnal freeze-thaw dynamics of a rock glacier and their impacts on mixing and solute transport"

_EGUsphere, 2024_

## Author Comment (AC1)

**CC1:** ['Comment on egusphere-2024-927'](), Giacomo Medici, 23 May 2024

General comments

Novel research in the field of hydrology. The manuscript needs some minor corrections that should improve the final version of the manuscript. See below the specific comments.

Author response (AR): We appreciate the positive feedback and the recognition of the novelty of our research. We will address each of the specific comments as detailed below.

Specific comments

Lines 14-68. The link between creep and groundwater flow is an unexplored topic. I would emphasize more this point in your introduction/discussion.

AR: We will include in the introduction and discussion sections a short review of knowledge on the link between rock glacier creep and groundwater flow. However, since it is not the main focus of the manuscript, we will keep the discussion minimal.

Lines 27-28. You mention snowmelt and groundwater flow in the introduction and the conceptual model. Please, expand this point and add recent literature on snowmelt aquifer recharge in mountain ranges that combines isotope analysis and monitoring:

- Lorenzi, V., Banzato, F., Barberio, M. D., Goeppert, N., Goldscheider, N., Gori, F., Lacchini A., Manetta M., Medici G., Rusi S., Petitta, M. 2024. Tracking flowpaths in a complex karst system through tracer test and hydrogeochemical monitoring: Implications for groundwater protection (Gran Sasso, Italy). Heliyon, 10(2).

- Stevenazzi, S., Zuffetti, C., Camera, C. A., Lucchelli, A., Beretta, G. P., Bersezio, R., & Masetti, M. (2023). Hydrogeological characteristics and water availability in the mountainous aquifer systems of Italian Central Alps: A regional scale approach. *Journal of Environmental Management*, *340*, 117958.

AR: We will expand the discussion on snowmelt and groundwater flow in both the introduction and the conceptual model sections. The references provided will be considered and included if suitable.

Line 68. Disclose the specific objectives of your research by using numbers (e.g., i, ii and iii) at the end of your introduction.

AR: Thanks for this suggestion. We will consider reformatting the introduction to better highlight the specific objectives.

Line 73. "Mostly". Please, specify the other lithologies. Alternatively, you can also fix the issue by deleting the vague term "mostly".

AR: Thanks. We will provide a more exhaustive description of the geology with the other lithologies present in the study area.

Line 77. "fractured aquifer". Insert more detail on the nature of the tectonic structures and joints. Thrusts and folds? Also normal faults? Unclear the nature of the fault zone in the conceptual model.

AR: Thanks for this comment. We agree that some information on the tectonics and nature of the fracture network would strengthen the manuscript. We will include this information in the new manuscript.

Line 280. Specify the area of the French Alps and the lithologies of the fractured bedrock aquifer there. Crystalline basement there?

AR: We will specify the area of the Swiss Alps being referred to and detailed the lithologies of the fractured bedrock aquifer. Basement is indeed not an appropriate term here.

Lines 397-550. Take into account the literature suggested above.

AR: We will consider the references provided and include them if appropriate.

Figures and tables

Figure 2. Insert an approximate spatial scale.

AR: Thanks for spotting this mistake. We will add the appropriate scale to figure 2.

Figure 6a. Do you need to add an equation and parameters ($R^2$) to the line?

AR: The line represents the local meteoric water line. No need to add an equation here, but we will add the label "LMWL" to the line to avoid any further confusion. Thanks.

Figure 8. Please, add the intermediate months on the horizontal axis.

AR: We will consider this suggestion and include the intermediate months if this does not overload the final figure.

Figure 10. Insert the spatial scale and specify if there is vertical exaggeration.

AR: the conceptual models are not to scale. We will include information in the figure caption to avoid confusion.

Figure 10. Unclear the nature of the fault zone. Normal fault, or thrust with vertical exaggeration? This point is unclear even by reading the text.

AR: We have no clear evidence of the existence and nature of the fault. Since our aim is to show a conceptualisation of the processes, rather than a site conceptual model, we will remove the fault to avoid confusion.

Figure 10 vs. Study Area and instrumentation. You need to provide more detail on the tectonic structures on the paragraph 2 to make clear the final conceptual model.

AR: Additional details on the tectonic structures will be provided in the second paragraph of the study area section.

---

## Author Comment (AC2)

**RC1:** ['Comment on egusphere-2024-927'](), Anonymous Referee #1, 13 Jun 2024

**General Comments:**

The authors present an interesting study of the Canfinal Rock Glacier in the Swiss Alps. They investigate the factors that have contributed to the rock glacier's flow and how the rock glacier contributes to spring flow during different times of year. The study was interesting, novel and I appreciate the use of multiple methods/lines of evidence to characterize the hydrological dynamics.

**Author response (AR):** Thank you for your positive feedback on the novelty of this work and for the constructive suggestions. We will address each of the comments.

My suggestions for improvement mostly focus on increasing context and detail in places. For the hydrochemistry results, more graphical representation of the hydrochemical signatures of different sample types and locations is needed. They are currently presented in PCA form in Figure 6 but the different waters (springs, streams, etc.) are not differentiated. Were any samples taken from the well? That would be interesting to see in comparison to the other samples. Consider adding an additional panel to Figure 6 to show EC versus some other ion with the sample types coloured or shaped by water type (spring, groundwater, stream, etc.) or location. Or else in panel B, you could make the points different shapes for different types of samples (although that may get too busy and be less clear). That way, the reader can easily see how the different samples and presumed end members relate to each other.

**AR:** Thank you for the insightful suggestions to improve the clarity and detail of the hydrochemistry results. We will enhance the graphical representation of the hydrochemical signatures in the future manuscript. We will make sure to update the Figures to differentiate the different water types (springs, streams, groundwater) by using distinct colours or shapes. We will also consider adding additional figures with scatter plots showing the relationship between the ions and electrical conductivities. Unfortunately, the wells are currently grouted, and the groundwater cannot be accessed for sampling.

In general, some additional explanation of the frequency-domain analysis methods and results would be helpful for readers who are not particularly familiar with these techniques. Around line 133, a conceptual statement about how the time-domain analysis is going to be interpreted would improve clarity. Consider adding a diagram to help clarify the phase shifting described on lines 148-150. In the results, the frequency-domain analysis results are not always intuitive, so some additional contextualization in text or annotation of the plots would likely help readers follow.

AR: In the future manuscript, we will provide additional explanation and context for the frequency-domain analysis methods and results. Around line 133, we will add a conceptual statement about how the time-domain analysis will be interpreted. Additionally, we will consider adding a figure like the one below to clarify the phase shifting described on lines 148-150 and provide more contextualization in the results section.

[Figure]

**Specific Comments:**

39: Briefly identify the other sources of water released from rock glaciers other than meltwater.

**AR:** We will add a brief description of other potential sources of water released from rock glaciers, such as precipitation, snowmelt and groundwater.

51: The hypotheses are nicely presented. Perhaps you could clarify that these hypotheses are not mutually exclusive. I.e., "…several hypotheses (which are not mutually exclusive) have been proposed…" if that is the case.

AR: Thank you for this suggestion. We will indeed revise the text to clarify that the hypotheses presented are not exclusive.

63: Could you add a sentence as to why this is the case?

AR: We will consider removing this hypothesis taken from the literature, as it is not highly pertinent.

64: The objectives are stated in the last paragraph of the introduction, but they come after mentioning the methods and the word objective is not used. I suggest stating 2-3 numbered objectives for maximum clarity.

AR: the reviewer is right that the actual format of the introduction can lead to confusions. We will reformat it and clearly state 2-3 numbered objectives.

121: Some basic details around sampling/analytical procedure (e.g., bottles, preservation, analytical equipment) would be expected here.

**AR:** In the future manuscript, we will include details on the sampling and analytical procedures used.

124: At this point, it's not clear what the correlation analysis is used for. Some explanation of the bigger picture is needed.

AR: The statistical analysis is aimed to identify the main water end-members. We will clarify this in the text illustrated by the new scatter plot figures.

Figure 4: Neat figure!

AR: Thanks!

158-169: These data sources and methods should be included in the methods section. Are there any limitations associated with ERA5 performance in mountains that should be acknowledged?

AR: Indeed - thanks for this suggestion. We will move the description of data sources and methods to the methods section and add a description on the limitation of ERA 5. We will include an assessment of ERA5 with nearby weather stations.

Figure 7: There are blocks in the EC data for April for S1, August for S2, January to March for S3 where EC is jumping between a certain value and 0 many times. Does that represent some kind of sensor error? Or is the spring going dry and reactivating in quick succession? This should be explained in the text.

AR: Thanks for spotting this. Indeed, it is when the sensors are not immersed because the water level at the spring is too low. We will modify the figures accordingly. We will also consider removing S2 and S3 because they are not analysed in depth in this study.

Figure 10: I suggest adding a legend entry for the light grey geologic material since all others are labeled (seasonally frozen talus hosting perennial ice lenses?). Also, why does the hillslope-scale flow line have such a bend? I found the captions "freezing conditions" and "thawing conditions" a little unclear and suggest simply "winter" and "summer" might be more intuitive.

AR: Indeed - we will simplify the flow lines in the conceptual model and modify the labels.

**Technical Corrections:**

188-189: The phrasing of this sentence is awkward, consider re-phrasing.

AR: A modification of this sentence will be considered in the future manuscript.

201: Should be "…this sampling campaign…"

AR: Noted.

---

## Author Comment (AC3)

**RC2:** ['Comment on egusphere-2024-927'](), Anonymous Referee #2, 14 Jun 2024

GENERAL COMMENTS

The manuscript by C. Louis, L.J.S. Halloran, and C. Roques provides an interesting and for the rock glacier research community novel hydro-chemical characterization of the previously uninvestigated Canfinal rock glacier and its surrounding springs in the southeastern Swiss Alps. I discuss the manuscript along its three storylines: (1) long-term kinematics and its relations to selected climatic drivers, (2) seasonal hydro-chemical (electrical conductivity EC, stable isotopes, major ions) characterization of several springs below the rock glacier, and (3) diurnal frequency-domain analysis of the EC of the rock glacier outflow. Finally, I have some suggestions for Fig. 10.

**Author response (AR):** Thank you for your thorough review and positive feedback on our manuscript. We appreciate your detailed comments and suggestions for improvement. We will address each point as detailed below.

First, the kinematic investigations, limited to a multi-year time scale by the available imagery, are interesting and well in line of the observations of the Swiss Permos Monitoring Network. Perhaps sufficient for the hydrological storyline would be the delineation and rough characterization of the rock glacier material (ice content) via the kinematics (L258-265) in support of the morphological evidence of ice-rich permafrost occurrence. Due to the scale mismatch, the relations between multi-year climatic and kinematics trends are hard to connect to the seasonal to daily/hourly hydrological analysis, although links between hydrology and kinematics undoubtedly exist. Still, keep it in the manuscript since it gives clues on the thermal state and provides valuable baseline kinematic observations on a previously uninvestigated, unknown site.

**AR:** We appreciate your acknowledgment of the value of our kinematic investigations. We agree that a clear connection between multi-year kinematics and annual/seasonal hydrological dynamics cannot be made with the existing data, but we agree with you that both of these analyses have value and should remain in the manuscript. Delineation and characterization of the rock glacier material does not appear feasible with a high level of confidence due to the lack of internal measurements in the RG. We will, however, elaborate our hypotheses on the evolution of ice content in the RG over the period of the available historic imagery.

Second, the seasonal hydro-chemical characterization enabled the seasonal differentiation of water sources contributing to the springs throughout the catchment. This was all very convincing and relevant. The spatio-temporal clustering of surface water chemistry (Figs. 6, S2) at small catchment scale reveals the distinct chemistry and "imprint" of the rock glacier compared to the vegetated plain "La Casina". The Canfinal borehole provides unique insights into the snow and permafrost interactions with groundwater in a thermally sensitive environment close to the lower permafrost limit. Concretely, Fig. 9 shows a link between ground water storage changes (trend of hydraulic head KB4) and EC S1, itself related to precipitation events and time elapsed since snowmelt.

AR: We appreciate your positive assessment on the hydrochemical data and interpretation.

Third, the diurnal frequency-domain analysis of EC is potentially a useful tool in shedding light on the timing of water and heat transfer and applied to the first time in alpine permafrost (to my knowledge). This is an important contribution. I think however that far-reaching interpretations on freeze/thaw cycles based on minuscule

0.5–2 µS/cm oscillations of the EC signal are presented too boldly: Low EC is associated with high discharge (L322) and linked to intense ground ice melt (L324) supposedly driven by diurnal temperature oscillations. Such a behavior might be more typical for (debris-covered) glaciers, but it is not typical for permafrost and rock glaciers. For the lack of independent, local measurements to corroborate these links, this remains a hypothesis and should be framed more cautiously. Please address:

AR: We appreciate your insights into the diurnal frequency-domain analysis. We agree that the text need to be revised to frame the interpretations on freeze/thaw cycles more cautiously, acknowledging the limitations of our interpretations and hypothesis. We will also provide more detail about the frequency-domain method in the SI.

1. No discharge observations are presented to corroborate the presumed (admittedly common) negative EC-discharge relation with discharge maxima in the afternoon. I am aware of the difficulties of obtaining a water level–discharge relation in such terrain. Given your repeated field visits: Do you have water level measurements/visual observations that would attest the afternoon high-flow at least qualitatively?

   AR: Unfortunately, it was not possible to measure discharge accurately in this diffuse system of springs. However, in the SI, if needed, we will provide evidence of discharge dynamics by providing water levels measured at the springs.

2. The assumption of low-EC, "clean" ground ice whose melt dilutes the outflow is untested on the Canfinal rock glacier. I cannot require you to dig a sample, but it should be mentioned that ground ice in rock glaciers was found to have differing solute content. The (few) available measurements of the chemical composition of rock glacier ice (exposures, drillcores) range from low-EC ice (e.g., Murtèl, Haeberli (ed.), 1990) that would result in dilution to "dirty" high-EC ice (e.g., Lazaun, Nickus et al., 2023) that would result in solute enrichment (Brighenti et al., 2021; Bearzot et al., 2023). On top of that, in a degrading rock glacier, two types of ground ice melt in the thaw season: First the 'young' ice in the active layer ('superimposed ice') and later the 'old' permafrost ice (del Siro et al., 2023).

   AR: This is a very interesting comment with very valuable references, thank you. As the reviewer mentioned, we unfortunately cannot sample the ice in the canfinal rock glacier. In the future manuscript, we will make sure to frame the interpretation cautiously and discuss the different hypotheses.

3. Please mention that dilution from ice melt is not the only behavior found in the outflow of rock glaciers. EC is also not necessarily a conservative tracer that solely hints at water provenance (ice melt) in periglacial/permafrost environments. Colombo et al. (2018) lists contrasting mechanisms of solute export and EC-discharge relations, some are regular, some are tied to precipitation events or weather spells, or related to weathering (enrichment, dilution, flushing). Briefly discuss which processes you consider likely given the measured regular EC oscillations.

   AR: Thank you again for this valuable comment. We will make sure to mention the changes in EC to potential processes, specifically related to weathering or sediment export.

4. No ground thermal data is shown to corroborate the diurnal freeze/thaw cycles. Due to the thermal inertia of the active layer, I strongly doubt that temperatures and melt rates at depths typical for ground ice melt in rock glaciers significantly vary on an hourly basis (certainly in the late thaw

season when the ground ice table has receded to depth)! The melting ice must be at or near the ground surface, not deeper than a few tenths of centimeters (the penetration length-scale of diurnal oscillations). Could you get an idea of the active layer thickness on Canfinal? This reasoning rather hints at snow or shallow seasonal ice hidden in the rough terrain – on the rock glacier but also on the adjacent talus slopes and headwalls. This explanation would be consistent with seasonally (broadly) decreasing amplitudes of the S1 diurnal cycles (Fig. 8): waning influence of snowmelt. Nonetheless, the pattern of discharge inversely varying with EC and concomitant with peak air temperatures is also reported by Mateo & Daniels (2018).

AR: Fully agree - we lack plenty of key data for the Canfinal rock Glacier which strongly limits the interpretation. Nevertheless, we will address the lack of ground thermal data and discuss the thermal inertia of the active layer in our revised manuscript. We will consider the possibility of snow or shallow seasonal ice as the source of diurnal EC oscillations and discuss the implications of this for our findings. The possibility of estimating the active layer thickness at Canfinal in the timeframe of this manuscript since ambitious. It will be the subject of future research on the site.

My point is: Your hypothesis is just one chain of processes out of many thinkable ones! Considering the complexity, it is not possible to make all links robust in the scope of this publication. My suggestion is that you introduce the novel diurnal frequency-domain analysis more cautiously as a tool and frame your ice melt-dilution hypothesis as one example of the chains of processes that can be explored with this tool. EC is a commonly measured variable. Many past & future data sets can be analyzed!

AR: Thanks - We appreciate this perspective and will introduce the diurnal frequency-domain analysis more cautiously in our revised manuscript.

Finally, Figure 10, the conceptual model sketch of the annual freeze-thaw cycles and its implications on groundwater flow. It is the first rock glacier hydrological model that focuses explicitly on their role in the entire catchments and brings up the permafrost interactions with deep groundwater flows. This is an important contribution. With a few modifications, permafrost and thermal aspects can be depicted more accurately, namely:

1. The extent of permafrost: The rock glacier, as a permafrost landform, is also in summer frozen (cryogenic, $\leq 0°C$), hence must be enclosed by the $0°C$ isotherm in both panels. Vice versa for the 'unfrozen till layer'.
2. Time/spatial scales of freeze/thawing: Only the active layer, the uppermost ca. 3-10 m beneath the surface, is subject to annual freezing/thawing. Thermal changes at depth are slow. A pervasive freezing/thawing of the bedrock with large changes at depth as depicted is not possible on a seasonal scale, the sketch rather evokes a long-term (decadal) permafrost degradation. Also, adding a spatial scale would help to grasp the spatio-temporal changes.
3. Site specificity: At the relatively low-altitude Canfinal site, available permafrost distribution maps (Map of potential permafrost distribution (Federal Office for the Environment FOEN) and the SLF 'Permafrost and ground ice map'; https://www.slf.ch/en/services-and-products/permafrost-and-ground-ice-map/) concordantly hint at patchy and likely shallow permafrost in the headwall that is not necessarily connected to the permafrost bound to the rock glacier below.

AR: Thank you for your positive assessment of the value of Figure 10 and for your detailed suggestions to improve it. All suggestions are highly relevant and will be included in the next version of the manuscript.

I suggest that the authors reshape the manuscript and resubmit it. I emphasize that the seasonal hydro-chemical characterization based on your large data set of sampled springs and the connection to the piezometer borehole is convincing. The frequency-domain analysis has its merits as a tool, there is no need to overstretch to explanations that are insufficiently backed up by local measurements. I am looking forward to receiving an updated version of the work!

SPECIFIC COMMENTS

L70ff (study site). Is the catchment currently glacierized or not? What is the mean annual air temperature and annual precipitation?

**AR:** The study site is currently not glacierized. We will provide the mean annual air temperature and annual precipitation in the site description.

L101ff (methodology). Snow cover duration: The determination of the snow cover duration, given its important role in the analysis, merits a few sentences in the methods section (currently only mentioned on L165). How reliable is the ERA5 snow cover product for complex terrain? To what extent might long-lasting snow among the coarse blocks on the rock glacier surface contribute to melt (Bearzot et al., 2023)?

**AR:** We will add more details on snow cover duration estimates in the methods section. We will also discuss the reliability of the ERA5 snow cover product for complex terrain and mention the potential contribution of long-lasting snow among the coarse blocks on the rock glacier surface to melt. However, this will remain descriptive as we don't have quantitative estimates/monitoring of snow cover in the catchments. This will be the scope of future research.

L118. When/at which intervals were the five sampling campaigns carried out?

**AR:** We will specify the timing and intervals of the five sampling campaigns in the methods section to provide clarity on the sampling schedule.

L134. The daily EC amplitudes the frequency analysis is based on are small within 0.5–2 µS/cm (Fig. 9A). No rock glacier study (to my knowledge) has harnessed EC data down to such fine resolution. How does this compare to the precision & resolution of the EC probes? The EC is weakly sensitive to water temperature. How was the measured EC corrected to 25°C? Were the EC loggers fully submerged also at low flow (or shaded/enclosed in a stilling well?) and water temperature reliably measured? Since I am not familiar with this analysis, this is intended as a request for clarification and not a critique.

AR: The EC probes have onboard temperature sensors for the temperature correction of EC. Thus, the analysed values are corrected to 25°C. The loggers were periodically exposed to air during low- and no-flow periods. These data were removed for the frequency-domain analyses. However, absolute values are not relevant when analysing temporal variations in the frequency-domain. Measurement noise in SC (see figure below) is clearly significantly less than the 1 cpd EC variations. We will clarify these points in the text.

L136 and Fig. 9. Especially towards the end of the thaw season, the EC signal is quite irregular and far from sinusoidal. How well does the 1-cpd component describe such a signal with a broader frequency band in terms of amplitude and phase? Showing the 1-cpd component would be helpful to grasp the method.

AR: Fourier analysis enables the isolation of the 1 cpd amplitude and phase, thus eliminating higher frequency "noise" and longer-term trends. Spectral leakage is still, nonetheless, possible. We have chosen the 3-day window as a good balance for minimizing this leakage, utilising the available data (rainfall-influence and dry periods are

excluded), and retaining adequate temporal resolution. As can be seen in figures 8 and 9, our approach reveals clear seasonal trends, but shorter-term variations cannot be interpreted. In our opinion, adding a subfigure to Figure 8 or 9 showing example frequency-domain data may add more confusion than clarity, but the inclusion of a figure in the SI appears suitable. For example:

[Figure]

L200. The PCA analysis is very intriguing! Fig. S2 is based on the Oct 2022 samples. How persistent is the found clustering over the season? This is shown in Fig. 6B but could be stated more clearly.

AR: We will look closely into the seasonal variations and include a comment on this in the manuscript. The October survey does, however, show the clearest distinction between end-members. This is expected as, for this autumn survey, snowmelt and precipitation.

L235, L239, L242. Measured facts (diurnal EC variations) are alongside interpretations (dilution behavior, melt driver). Please move the latter to the discussion Sect. 5.3 to avoid repetition (i.e., "…seasonal trends which, for the snow-free period, can be interpreted as measures of the intensity of dilution from RG melt", "The ratio of the EC 1 cpd amplitudes to those of Tair normalizes the EC amplitudes by the main driver of daily melt rate variations", "…indicating a potentially significant contribution from RG meltwater").

AR: We thank you for this comment and agree that parts of this text are better suited to the discussion section.

L301-304: "An isolated contribution from the Canfinal RG cannot be detected…" This important finding is furthermore corroborated by Fig. S2 (PCA, spatial coherence): The distinct geochemical signal is lost a few hundred meters downstream of the rock glacier front. Please add Bearzot et al. (2023) at L304 as they also provided an estimate.

AR: Thanks - The dominant contribution of groundwater at the talus that progressively hides the signal of the rock glacier is an important outcome of this work for the understanding of the hydrology of the site. We will add the interesting reference of Bearzot et al. (2023) to support this has been observed in other contexts.

L290-318: Very interesting!

AR: Thanks!

L312. "Some suggest that bacterial activity…" Who?

AR: We thought about this contribution and will mention the relevant literature in the final manuscript.

L332. Please write "the active layer thickens" instead of "the ice thickness in the active layer decreases".

Fig. 5. A neat figure!

AR: Thanks! Complement the comment from RC1 regarding figure 4 ;).

Fig. 6. The single most important figure, panel B could be enlarged. Same color coding of the months in panels A and B eases comparison, nice! The ellipses in B) are unnecessary, the different coloring distracts. What do the different circle sizes mean? Could a few key springs (among S1) be marked so that we can follow how their chemistry evolves in the PC plot?

AR: Interesting suggestions. We will make sure to include them in the revised version of this figure.

Fig. 7, caption. Should read "July 2022", not "July 2002". Just a thought: Flipping the map or the order of the EC panels would place the data next to location in the map, the more so, as the labels of the EC data set are "hidden" in the subscript of the y-axis label (optional).

AR: You are right - thanks for these comments!

Fig. 9A. What exactly means 'filtered' here (cleaned?) and why is the S1 EC here in the range 100–130 μS/cm whereas it is 50–75 μS/cm in Fig. 7? Am I missing something?

AR: We will modify this figure for clarity and add detail to the caption. Regarding the difference between Fig 9A and Fig 7, it was a mistake as non-temperature corrected data was accidentally included in the figure.

REFERENCES

Bearzot, F., Colombo, N., Cremonese, E., di Cella, U. M., Drigo, E., Caschetto, M., ... & Rossini, M. (2023). Hydrological, thermal and chemical influence of an intact rock glacier discharge on mountain stream water. Science of The Total Environment, 876, 162777.

Brighenti, S., Engel, M., Tolotti, M., Bruno, M. C., Wharton, G., Comiti, F., ... & Bertoldi, W. (2021). Contrasting physical and chemical conditions of two rock glacier springs. Hydrological Processes, 35(4), e14159.

Colombo, N., Gruber, S., Martin, M., Malandrino, M., Magnani, A., Godone, D., ... & Salerno, F. (2018). Rainfall as primary driver of discharge and solute export from rock glaciers: The Col d'Olen Rock Glacier in the NW Italian Alps. Science of the Total Environment, 639, 316-330.

Del Siro, C., Scapozza, C., Perga, M. E., & Lambiel, C. (2023). Investigating the origin of solutes in rock glacier springs in the Swiss Alps: A conceptual model. Frontiers in Earth Science, 11, 1056305.

Haeberli, W., ed.: Pilot analysis of permafrost cores from the active rock glacier Murtèl I, Piz Corvatsch, Eastern Swiss Alps. A workshop report., no. 9 in Arbeitsheft, VAW/ETH Zürich, 1990.

Mateo, E. I., & Daniels, J. M. (2019). Surface hydrological processes of rock glaciated basins in the San Juan Mountains, Colorado. Physical Geography, 40(3), 275-293.

Nickus, U., Thies, H., Krainer, K., Lang, K., Mair, V., & Tonidandel, D. (2023). A multi-millennial record of rock glacier ice chemistry (Lazaun, Italy). Frontiers in Earth Science, 11, 1141379.

AR: Thank you for these references.

---

## Author Response (AR1)

**Response to reviewers for "Seasonal and diurnal freeze-thaw dynamics of a rock glacier and their impacts on mixing and solute transport"**
**by Cyprien Louis, Landon J. S. Halloran, and Clément Roques**

**To Editor Prof. Ali,**

**We appreciate your handling of our manuscript and the constructive feedbacks provided by the reviewers. Their comments and suggestions have allowed us to improve the clarity and focus of our paper.**

**In this revised version, we have addressed all the reviewers' comments and detailed our responses below, highlighted in blue. Additionally, we have enriched our analysis by incorporating more recent data gathered during the 2023/2024 field campaigns. Specifically, we have expanded the dataset with updated air temperature and electrical conductivity time series from the main springs, which have been included in the frequency-domain analysis. While the overarching interpretations remain consistent, these additions strengthen the paper's outcomes and reinforce our conclusions.**

**Furthermore, we have streamlined the manuscript by transferring certain figures that are less central to our interpretation to the supplementary material. The aim is to enhance the paper's focus.**

**We hope that the revised manuscript meets the expectations of the reviewers and will be considered for publication.**

**Best regards,**

**The authors**
* * *
**CC1:** ['Comment on egusphere-2024-927'](), Giacomo Medici, 23 May 2024

General comments

Novel research in the field of hydrology. The manuscript needs some minor corrections that should improve the final version of the manuscript. See below the specific comments.

Author response (AR): We appreciate the positive feedback and the recognition of the novelty of our research. We have addressed each of the specific comments as detailed below.

Specific comments

Lines 14-68. The link between creep and groundwater flow is an unexplored topic. I would emphasize more this point in your introduction/discussion.

AR: In this new version, we have completed the introduction and discussion sections to include a background knowledge on the link between rock glacier creep and groundwater flow. We have added a sentence in the introduction in lines 32-35.

Lines 27-28. You mention snowmelt and groundwater flow in the introduction and the conceptual model. Please, expand this point and add recent literature on snowmelt aquifer recharge in mountain ranges that combines isotope analysis and monitoring:

- Lorenzi, V., Banzato, F., Barberio, M. D., Goeppert, N., Goldscheider, N., Gori, F., Lacchini A., Manetta M., Medici G., Rusi S., Petitta, M. 2024. Tracking flowpaths

in a complex karst system through tracer test and hydrogeochemical monitoring: Implications for groundwater protection (Gran Sasso, Italy). Heliyon, 10(2).

- Stevenazzi, S., Zuffetti, C., Camera, C. A., Lucchelli, A., Beretta, G. P., Bersezio, R., & Masetti, M. (2023). Hydrogeological characteristics and water availability in the mountainous aquifer systems of Italian Central Alps: A regional scale approach. *Journal of Environmental Management*, *340*, 117958.

AR: In this new version of the manuscript, we aimed at providing more discussion on snowmelt and groundwater flow in both the introduction and the conceptual model sections. The references provided has been considered and included if judge suitable.

Line 68. Disclose the specific objectives of your research by using numbers (e.g., i, ii and iii) at the end of your introduction.

AR: We have clarified our specific objectives at the end of the introduction.

Line 73. "Mostly". Please, specify the other lithologies. Alternatively, you can also fix the issue by deleting the vague term "mostly".

AR: We now have provided a more exhaustive description of the geological and tectonic settings.

Line 77. "fractured aquifer". Insert more detail on the nature of the tectonic structures and joints. Thrusts and folds? Also normal faults? Unclear the nature of the fault zone in the conceptual model.

AR: We have added information on the geology, tectonic setting and on the nature of the fractures/faults in the section 2.

Line 280. Specify the area of the French Alps and the lithologies of the fractured bedrock aquifer there. Crystalline basement there?

AR: refer to previous answer.

Lines 397-550. Take into account the literature suggested above.

AR: References provided by all reviewers have been considered and included if appropriate.

Figures and tables

Figure 2. Insert an approximate spatial scale.

AR: Adding a scale on the picture is not relevant here. However, we have added a sentence in the caption: "For reference, the front of the rock glacier is approximately 250 meters wide".

Figure 6a. Do you need to add an equation and parameters (R2) to the line?

AR: The line represents the local meteoric water line. No need to add an equation on the figure as it is mentioned in the text. We have added the label "LMWL" in the legend to avoid any further confusion.

Figure 8. Please, add the intermediate months on the horizontal axis.

AR: This figure has been completely redone, with S1 and S4 fully separated into subfugures and additional data (2023) added. Including every month in the x-axis tick labels is not relevant.

Figure 10. Insert the spatial scale and specify if there is vertical exaggeration.

AR: the conceptual models are not to scale. We included the sentence in the figure caption.

Figure 10. Unclear the nature of the fault zone. Normal fault, or thrust with vertical exaggeration? This point is unclear even by reading the text.

AR: We have no clear evidence of the existence and nature of the fault. Since our aim is to show a conceptualisation of the processes, rather than a site conceptual model with actual tectonic settings, we have removed the fault to avoid confusion.

Figure 10 vs. Study Area and instrumentation. You need to provide more detail on the tectonic structures on the paragraph 2 to make clear the final conceptual model.

AR: Additional details on the tectonic structures is now provided in section 2.
* * *
RC1: ['Comment on egusphere-2024-927'](), Anonymous Referee #1, 13 Jun 2024

**General Comments:**

The authors present an interesting study of the Canfinal Rock Glacier in the Swiss Alps. They investigate the factors that have contributed to the rock glacier's flow and how the rock glacier contributes to spring flow during different times of year. The study was interesting, novel and I appreciate the use of multiple methods/lines of evidence to characterize the hydrological dynamics.

AR: Thank you for your positive feedback on the novelty of this work and for the constructive suggestions.

My suggestions for improvement mostly focus on increasing context and detail in places. For the hydrochemistry results, more graphical representation of the hydrochemical signatures of different sample types and locations is needed. They are currently presented in PCA form in Figure 6 but the different waters (springs, streams, etc.) are not differentiated. Were any samples taken from the well? That would be interesting to see in comparison to the other samples. Consider adding an additional panel to Figure 6 to show EC versus some other ion with the sample types coloured or shaped by water type (spring, groundwater, stream, etc.) or location. Or else in panel B, you could make the points different shapes for different types of samples (although that may get too busy and be less clear). That way, the reader can easily see how the different samples and presumed end members relate to each other.

AR: Thank you for the insightful suggestions to improve the clarity and detail of the hydrochemistry results. We have now enhanced the graphical representation of the hydrochemical signatures in the future manuscript. Figure 6 is now new and highlight better the 2 end-members (RG-influenced vs deep groundwater) and their temporal dynamics. We added additional figures with scatter plots showing the relationship between the major ions. The section 4.2 focusing on the hydrochemical interpretation has been improved.

Unfortunately, the wells are grouted, and the groundwater cannot be accessed for sampling.

In general, some additional explanation of the frequency-domain analysis methods and results would be helpful for readers who are not particularly familiar with these techniques. Around line 133, a conceptual statement about how the time-domain analysis is going to be interpreted would improve clarity. Consider adding a diagram to help clarify the phase shifting described on lines 148-150. In the results, the frequency-domain analysis results are not always intuitive, so some additional contextualization in text or annotation of the plots would likely help readers follow.

In the new version of the manuscript, we now provide additional explanation and context for the frequency-domain analysis methods and results. A figure like that describes the phase shifting is available in supplementary material. We have extensively expanded the interpretation discussion in Section 5.3.

**Specific Comments:**

39: Briefly identify the other sources of water released from rock glaciers other than meltwater.

**AR:** We have completed the introduction that provides now a more exhaustive review on the potential sources of water released from rock glaciers (lines 52-63).

51: The hypotheses are nicely presented. Perhaps you could clarify that these hypotheses are not mutually exclusive. I.e., "…several hypotheses (which are not mutually exclusive) have been proposed…" if that is the case.

AR: Thank you for this suggestion. We have indeed revised the text to clarify that the hypotheses presented are not exclusive (line 32).

63: Could you add a sentence as to why this is the case?

AR: We have removed this hypothesis, as it is not highly pertinent for our study.

64: The objectives are stated in the last paragraph of the introduction, but they come after mentioning the methods and the word objective is not used. I suggest stating 2-3 numbered objectives for maximum clarity.

AR: We have rephrased and clarify the objectives of our study as (lines 64-70): "In this study, we employ a multi-method approach to investigate the cryo-hydrogeological functioning of the Val d'Ursé Critical Zone Observatory in Switzerland, where a major rock glacier (the Canfinal rock glacier) is present. By combining digital image correlation, hydrochemical analysis, and a novel frequency-domain analysis of temperature and electrical conductivity monitored at spring locations, we aim to resolve the seasonal to diurnal freeze-thaw dynamics of the RG, identify multi-year trends in its creep rates and their meteorological drivers, and elucidate cryosphere-groundwater interactions. This comprehensive approach not only enhances our understanding of the complex hydrological processes in the Val d'Ursé headwater catchment but also generates knowledge that can be applied to other alpine regions with similar environmental settings."

121: Some basic details around sampling/analytical procedure (e.g., bottles, preservation, analytical equipment) would be expected here.

**AR:** We have now included more details on the sampling and analytical procedures used.

124: At this point, it's not clear what the correlation analysis is used for. Some explanation of the bigger picture is needed.

AR: We agree that the correlation analysis was not clearly used in the first version of the manuscript. In this newer version, we have removed this analysis.

Figure 4: Neat figure!

AR: Thanks!

158-169: These data sources and methods should be included in the methods section. Are there any limitations associated with ERA5 performance in mountains that should be acknowledged?

AR: Indeed - thanks for this suggestion. We have moved the description of data sources and methods to the methods section and added description on the limitation of ERA5 (line 183).

Figure 7: There are blocks in the EC data for April for S1, August for S2, January to March for S3 where EC is jumping between a certain value and 0 many times. Does that represent some kind of sensor error? Or is the spring going dry and reactivating in quick succession? This should be explained in the text.

AR: Thanks for spotting this. Indeed, it is when the sensors are not immersed because the water level at the spring is too low. We have now removed the time series of S2 and S3 because they are not analysed in depth in this study and added a comment on the data noise and gap in the caption of the Figure 7.

Figure 10: I suggest adding a legend entry for the light grey geologic material since all others are labeled (seasonally frozen talus hosting perennial ice lenses?). Also, why does the hillslope-scale flow line have such a bend? I found the captions "freezing conditions" and "thawing conditions" a little unclear and suggest simply "winter" and "summer" might be more intuitive.

AR: We have simplified the flow lines in the conceptual model. We decided to keep the labels as it was to make a clear link with the objective of the study which focuses on analysing the freeze-taw cycle of the RG and its influence on the hydrogeological system.

**Technical Corrections:**

188-189: The phrasing of this sentence is awkward, consider re-phrasing.

AR: The first 2 paragraphs of Section 4.2 have been rewritten.

201: Should be "…this sampling campaign…"

AR: modified.
* * *
RC2: 'Comment on egusphere-2024-927', Anonymous Referee #2, 14 Jun 2024

GENERAL COMMENTS

The manuscript by C. Louis, L.J.S. Halloran, and C. Roques provides an interesting and for the rock glacier research community novel hydro-chemical characterization of the previously uninvestigated Canfinal rock glacier and its surrounding springs in the southeastern Swiss Alps. I discuss the manuscript along its three storylines: (1) long-term kinematics and its relations to selected climatic drivers, (2) seasonal

hydro-chemical (electrical conductivity EC, stable isotopes, major ions) characterization of several springs below the rock glacier, and (3) diurnal frequency-domain analysis of the EC of the rock glacier outflow. Finally, I have some suggestions for Fig. 10.

**AR:** Thank you for your thorough review and positive feedback on our manuscript. We appreciate your detailed comments and suggestions for improvement. We believe that the modifications provided in response to the comments raised have greatly strengthen the manuscript. Thanks.

First, the kinematic investigations, limited to a multi-year time scale by the available imagery, are interesting and well in line of the observations of the Swiss Permos Monitoring Network. Perhaps sufficient for the hydrological storyline would be the delineation and rough characterization of the rock glacier material (ice content) via the kinematics (L258-265) in support of the morphological evidence of ice-rich permafrost occurrence. Due to the scale mismatch, the relations between multi-year climatic and kinematics trends are hard to connect to the seasonal to daily/hourly hydrological analysis, although links between hydrology and kinematics undoubtedly exist. Still, keep it in the manuscript since it gives clues on the thermal state and provides valuable baseline kinematic observations on a previously uninvestigated, unknown site.

**AR:** We appreciate your acknowledgment of the value of our kinematic investigations. We agree that a clear connection between multi-year kinematics and annual/seasonal hydrological dynamics cannot be made with the existing data, but we agree with you that both of these analyses have value and should remain in the manuscript. Delineation and characterization of the rock glacier material does not appear feasible with a high level of confidence due to the lack of internal measurements in the RG. In this new version of the manuscript we have completed the discussion on the link between kinematics and hydrological analyses. We also elaborate our hypotheses on the evolution of ice content in the RG over the period of the available historic imagery.

Second, the seasonal hydro-chemical characterization enabled the seasonal differentiation of water sources contributing to the springs throughout the catchment. This was all very convincing and relevant. The spatio-temporal clustering of surface water chemistry (Figs. 6, S2) at small catchment scale reveals the distinct chemistry and "imprint" of the rock glacier compared to the vegetated plain "La Casina". The Canfinal borehole provides unique insights into the snow and permafrost interactions with groundwater in a thermally sensitive environment close to the lower permafrost limit. Concretely, Fig. 9 shows a link between ground water storage changes (trend of hydraulic head KB4) and EC S1, itself related to precipitation events and time elapsed since snowmelt.

AR: We appreciate your positive assessment.

Third, the diurnal frequency-domain analysis of EC is potentially a useful tool in shedding light on the timing of water and heat transfer and applied to the first time in alpine permafrost (to my knowledge). This is an important contribution. I think however that far-reaching interpretations on freeze/thaw cycles based on minuscule 0.5-2 µS/cm oscillations of the EC signal are presented too boldly: Low EC is associated with high discharge (L322) and linked to intense ground ice melt (L324) supposedly driven by diurnal temperature oscillations. Such a behavior might be more typical for (debris-covered) glaciers, but it is not typical for permafrost and rock glaciers. For the lack of independent, local measurements to corroborate these links, this remains a hypothesis and should be framed more cautiously. Please address:

AR: Following the above comment and the details in 4 points detailed below, we have extensively rewritten and added to Section 5.3. The section now is explicit in the assumptions (based, in part, on the absence of high-solubility rocks in the RG root zone), while reiterating that we are only proposing a hypothesis. We also better link to the published literature (thank you again for the references) and state the kinds of (future) investigations that would help us confirm or reject our hypothesis.

1. No discharge observations are presented to corroborate the presumed (admittedly common) negative EC-discharge relation with discharge maxima in the afternoon. I am aware of the difficulties of obtaining a water level-discharge relation in such terrain. Given your repeated field visits: Do you have water level measurements/visual observations that would attest the afternoon high-flow at least qualitatively?

   AR: Unfortunately, it was not possible to measure discharge accurately in this diffuse system of springs.

2. The assumption of low-EC, "clean" ground ice whose melt dilutes the outflow is untested on the Canfinal rock glacier. I cannot require you to dig a sample, but it should be mentioned that ground ice in rock glaciers was found to have differing solute content. The (few) available measurements of the chemical composition of rock glacier ice (exposures, drillcores) range from low-EC ice (e.g., Murtèl, Haeberli (ed.), 1990) that would result in dilution to "dirty" high-EC ice (e.g., Lazaun, Nickus et al., 2023) that would result in solute enrichment (Brighenti et al., 2021; Bearzot et al., 2023). On top of that, in a degrading rock glacier, two types of ground ice melt in the thaw season: First the 'young' ice in the active layer ('superimposed ice') and later the 'old' permafrost ice (del Siro et al., 2023).

   AR: This is a very interesting comment with very valuable references, thank you. As the reviewer mentioned, we unfortunately cannot sample the ice in the canfinal rock glacier. In this new version of the manuscript, we ensured to frame the interpretation cautiously and discuss the different hypotheses.

3. Please mention that dilution from ice melt is not the only behavior found in the outflow of rock glaciers. EC is also not necessarily a conservative tracer that solely hints at water provenance (ice melt) in periglacial/permafrost environments. Colombo et al. (2018) lists contrasting mechanisms of solute export and EC-discharge relations, some are regular, some are tied to precipitation events or weather spells, or related to weathering (enrichment, dilution, flushing). Briefly discuss which processes you consider likely given the measured regular EC oscillations.

   AR: We have modified the manuscript accordingly.

4. No ground thermal data is shown to corroborate the diurnal freeze/thaw cycles. Due to the thermal inertia of the active layer, I strongly doubt that temperatures and melt rates at depths typical for ground ice melt in rock glaciers significantly vary on an hourly basis (certainly in the late thaw season when the ground ice table has receded to depth)! The melting ice must be at or near the ground surface, not deeper than a few tenths of centimeters (the penetration length-scale of diurnal oscillations). Could you get an idea of the active layer thickness on Canfinal? This reasoning rather hints at snow or shallow seasonal ice hidden in the rough terrain – on the rock glacier but also on the adjacent talus slopes and headwalls. This explanation would be consistent with seasonally (broadly) decreasing amplitudes of the S1 diurnal cycles (Fig. 8): waning influence of snowmelt. Nonetheless, the

pattern of discharge inversely varying with EC and concomitant with peak air temperatures is also reported by Mateo & Daniels (2018).

> AR: Thank you for this comment. We agree that we lack key ground thermal data for the Canfinal rock glacier which strongly limits the interpretation on the freezing-thawing cycles at depth. We acknowledge this limitation in the discussion. We have strengthened the interpretation of the possibility of snow or shallow seasonal ice as the source of diurnal EC oscillations and discuss the implications of this for our findings. The possibility of estimating the active layer thickness at Canfinal in the timeframe of this manuscript is ambitious. It will be the subject of future research on the site.

My point is: Your hypothesis is just one chain of processes out of many thinkable ones! Considering the complexity, it is not possible to make all links robust in the scope of this publication. My suggestion is that you introduce the novel diurnal frequency-domain analysis more cautiously as a tool and frame your ice melt-dilution hypothesis as one example of the chains of processes that can be explored with this tool. EC is a commonly measured variable. Many past & future data sets can be analyzed!

AR: Thanks - We appreciate this perspective and now introduce the diurnal frequency-domain analysis more cautiously in our revised manuscript.

Finally, Figure 10, the conceptual model sketch of the annual freeze-thaw cycles and its implications on groundwater flow. It is the first rock glacier hydrological model that focuses explicitly on their role in the entire catchments and brings up the permafrost interactions with deep groundwater flows. This is an important contribution. With a few modifications, permafrost and thermal aspects can be depicted more accurately, namely:

1. The extent of permafrost: The rock glacier, as a permafrost landform, is also in summer frozen (cryogenic, ≤0°C), hence must be enclosed by the 0°C isotherm in both panels. Vice versa for the 'unfrozen till layer'.
2. Time/spatial scales of freeze/thawing: Only the active layer, the uppermost ca. 3–10 m beneath the surface, is subject to annual freezing/thawing. Thermal changes at depth are slow. A pervasive freezing/thawing of the bedrock with large changes at depth as depicted is not possible on a seasonal scale, the sketch rather evokes a long-term (decadal) permafrost degradation. Also, adding a spatial scale would help to grasp the spatio-temporal changes.
3. Site specificity: At the relatively low-altitude Canfinal site, available permafrost distribution maps (Map of potential permafrost distribution (Federal Office for the Environment FOEN) and the SLF 'Permafrost and ground ice map'; https://www.slf.ch/en/services-and-products/permafrost-and-ground-ice-map/) concordantly hint at patchy and likely shallow permafrost in the headwall that is not necessarily connected to the permafrost bound to the rock glacier below.

AR: Thank you for your positive assessment of the value of Figure 10 and for your detailed suggestions to improve it. All suggestions are highly relevant and have been included in the new version of the conceptual models.

I suggest that the authors reshape the manuscript and resubmit it. I emphasize that the seasonal hydro-chemical characterization based on your large data set of sampled springs and the connection to the piezometer borehole is convincing. The frequency-domain analysis has its merits as a tool, there is no need to overstretch to explanations that are insufficiently backed up by local measurements. I am looking forward to receiving an updated version of the work!

SPECIFIC COMMENTS

L70ff (study site). Is the catchment currently glacierized or not? What is the mean annual air temperature and annual precipitation?

AR: The study site is currently not glacierized. We have now provided the mean annual air temperature and annual precipitation in the site description (line 86).

L101ff (methodology). Snow cover duration: The determination of the snow cover duration, given its important role in the analysis, merits a few sentences in the methods section (currently only mentioned on L165). How reliable is the ERA5 snow cover product for complex terrain? To what extent might long-lasting snow among the coarse blocks on the rock glacier surface contribute to melt (Bearzot et al., 2023)?

AR: We added details on snow cover duration estimates in the methods section. We also mention the reliability of the ERA5 for complex terrain and mention the potential contribution of long-lasting snow among the coarse blocks on the rock glacier surface to melt. However, this remain descriptive as we don't have quantitative estimates/monitoring of snow cover in the catchments. This will be the scope of future research.

L118. When/at which intervals were the five sampling campaigns carried out?

AR: We now specify the dates of the sampling campaigns in the methods section (line 131).

L134. The daily EC amplitudes the frequency analysis is based on are small within 0.5–2 µS/cm (Fig. 9A). No rock glacier study (to my knowledge) has harnessed EC data down to such fine resolution. How does this compare to the precision & resolution of the EC probes? The EC is weakly sensitive to water temperature. How was the measured EC corrected to 25°C? Were the EC loggers fully submerged also at low flow (or shaded/enclosed in a stilling well?) and water temperature reliably measured? Since I am not familiar with this analysis, this is intended as a request for clarification and not a critique.

AR: The EC probes have onboard temperature sensors for the temperature correction of EC. Thus, the analysed values are corrected to 25°C. The loggers were periodically exposed to air during low- and no-flow periods. These data were removed for the frequency-domain analyses. However, absolute values are not relevant when analysing temporal variations in the frequency-domain. Measurement noise in SC (see figure below) is clearly significantly less than the 1 cpd EC variations. We have clarified all of this in Section 3.3 and have added a new SI figure (S2).

L136 and Fig. 9. Especially towards the end of the thaw season, the EC signal is quite irregular and far from sinusoidal. How well does the 1-cpd component describe such a signal with a broader frequency band in terms of amplitude and phase? Showing the 1-cpd component would be helpful to grasp the method.

AR: Fourier analysis enables the isolation of the 1 cpd amplitude and phase, thus eliminating higher frequency "noise" and longer-term trends. Spectral leakage is still, nonetheless, possible. We have chosen the 3-day window as a good balance for minimizing this leakage, utilising the available data (rainfall-influence and dry periods are excluded), and retaining adequate temporal resolution. Our approach reveals clear seasonal trends, but shorter-term variations cannot be interpreted. We show an example frequency-domain data in the SI.

L200. The PCA analysis is very intriguing! Fig. S2 is based on the Oct 2022 samples. How persistent is the found clustering over the season? This is shown in Fig. 6B but could be stated more clearly.

AR: In this new version of the manuscript, we have provided new figures for the interpretation of seasonal variability of the main endmembers. The October 2022 survey does, however, show the clearest distinction between end-members. This is expected as this survey was carried out during an extremely dry period.

L235, L239, L242. Measured facts (diurnal EC variations) are alongside interpretations (dilution behavior, melt driver). Please move the latter to the discussion Sect. 5.3 to avoid repetition (i.e., "…seasonal trends which, for the snow-free period, can be interpreted as measures of the intensity of dilution from RG melt", "The ratio of the EC 1 cpd amplitudes to those of Tair normalizes the EC amplitudes by the main driver of daily melt rate variations", "…indicating a potentially significant contribution from RG meltwater").

AR: We now have limited the interpretation language in Section 4.3 and refer to reader to the appropriate sections in the Discussion.

L301–304: "An isolated contribution from the Canfinal RG cannot be detected…" This important finding is furthermore corroborated by Fig. S2 (PCA, spatial coherence): The distinct geochemical signal is lost a few hundred meters downstream of the rock glacier front. Please add Bearzot et al. (2023) at L304 as they also provided an estimate.

AR: Thanks – The dominant contribution of groundwater at the talus that progressively hides the signal of the rock glacier is an important outcome of this work for the understanding of the hydrology of the site. We have emphasized this in the section 5.3 of the discussion. We also have included the reference.

L290–318: Very interesting!

AR: Thanks!

L312. "Some suggest that bacterial activity…" Who?

AR: We have added references to Tolotti et al. (2020) and Fegel et al. (2016).

L332. Please write "the active layer thickens" instead of "the ice thickness in the active layer decreases".

AR: Done.

Fig. 5. A neat figure!

AR: Thanks!

Fig. 6. The single most important figure, panel B could be enlarged. Same color coding of the months in panels A and B eases comparison, nice! The ellipses in B) are unnecessary, the different coloring distracts. What do the different circle sizes mean? Could a few key springs (among S1) be marked so that we can follow how their chemistry evolves in the PC plot?

AR: We have now modified the figure in panel D of Figure 6.

Fig. 7, caption. Should read "July 2022", not "July 2002". Just a thought: Flipping the map or the order of the EC panels would place the data next to location in the

map, the more so, as the labels of the EC data set are "hidden" in the subscript of the y-axis label (optional).

AR: We have provided a new figure 7.

Fig. 9A. What exactly means 'filtered' here (cleaned?) and why is the S1 EC here in the range 100–130 µS/cm whereas it is 50–75 µS/cm in Fig. 7? Am I missing something?

LH: The filtering is discussed in section 3.3. Regarding the issue of the 25C-corrected EC vs. raw EC, we decided to remove the former Figure 7 due to it not being essential for our study.

REFERENCES

Bearzot, F., Colombo, N., Cremonese, E., di Cella, U. M., Drigo, E., Caschetto, M., ... & Rossini, M. (2023). Hydrological, thermal and chemical influence of an intact rock glacier discharge on mountain stream water. Science of The Total Environment, 876, 162777.

Brighenti, S., Engel, M., Tolotti, M., Bruno, M. C., Wharton, G., Comiti, F., ... & Bertoldi, W. (2021). Contrasting physical and chemical conditions of two rock glacier springs. Hydrological Processes, 35(4), e14159.

Colombo, N., Gruber, S., Martin, M., Malandrino, M., Magnani, A., Godone, D., ... & Salerno, F. (2018). Rainfall as primary driver of discharge and solute export from rock glaciers: The Col d'Olen Rock Glacier in the NW Italian Alps. Science of the Total Environment, 639, 316-330.

Del Siro, C., Scapozza, C., Perga, M. E., & Lambiel, C. (2023). Investigating the origin of solutes in rock glacier springs in the Swiss Alps: A conceptual model. Frontiers in Earth Science, 11, 1056305.

Haeberli, W., ed.: Pilot analysis of permafrost cores from the active rock glacier Murtèl I, Piz Corvatsch, Eastern Swiss Alps. A workshop report., no. 9 in Arbeitsheft, VAW/ETH Zürich, 1990.

Mateo, E. I., & Daniels, J. M. (2019). Surface hydrological processes of rock glaciated basins in the San Juan Mountains, Colorado. Physical Geography, 40(3), 275-293.

Nickus, U., Thies, H., Krainer, K., Lang, K., Mair, V., & Tonidandel, D. (2023). A multi-millennial record of rock glacier ice chemistry (Lazaun, Italy). Frontiers in Earth Science, 11, 1141379.